# Infinite Mask Diffusion for Few-Step Distillation

**Jaehoon Yoo** [* 1]   **Wonjung Kim** [* 1]   **Chanhyuk Lee** [1]   **Seunghoon Hong** [1]

## Abstract

Masked Diffusion Models (MDMs) have emerged as a promising alternative to autoregressive models in language modeling, offering the advantages of parallel decoding and bidirectional context processing within a simple yet effective framework. Specifically, their explicit distinction between masked tokens and data underlies their simple framework and effective conditional generation. However, MDMs typically require many sampling iterations due to factorization errors stemming from simultaneous token updates. We observe that a theoretical lower bound of the factorization error exists, which standard MDMs cannot reduce due to their use of a deterministic single-state mask. In this paper, we propose the Infinite Mask Diffusion Model (IMDM), which introduces a stochastic infinite-state mask to mitigate the theoretical bound while directly inheriting the benefits of MDMs, including the compatibility with pre-trained weights. We empirically demonstrate that MDM fails to perform few-step generation even in a simple synthetic task due to the factorization error bound, whereas IMDM can find an efficient solution for the same task. Finally, when equipped with appropriate distillation methods, IMDM surpasses existing few-step distillation methods at small step counts on LM1B and OpenWebText.[1]

## 1. Introduction

Recently, Masked Diffusion Models (MDMs) (Sahoo et al., 2024; Shi et al., 2024; Ou et al., 2025) have gained significant traction as a promising alternative to Autoregressive (AR) models (Radford et al., 2019; Grattafiori et al.,

---
[*]Equal contribution  [1]Korea Advanced Institute of Science and Technology (KAIST). Correspondence to: Jaehoon Yoo <wogns98@kaist.ac.kr>, Wonjung Kim <wjhj16@kaist.ac.kr>, Seunghoon Hong <seunghoon.hong@kaist.ac.kr>.

*Proceedings of the 43$^{rd}$ International Conference on Machine Learning*, Seoul, South Korea. PMLR 306, 2026. Copyright 2026 by the author(s).

[1]Project Page: `Ugness.github.io/official_imdm`

2024). Specifically, MDMs are characterized by their capability for bidirectional and parallel decoding, offering a distinct advantage over the sequential nature of AR models. Driven by these advantages, active research is underway to utilize MDMs as a backbone for Large Language Models (LLMs) (Nie et al., 2025; Labs et al., 2025; Song et al., 2025). MDMs are trained on simple masked token prediction objectives similar to BERT (Devlin et al., 2019), enabling them to perform conditional generative downstream tasks seamlessly from an unconditional model, such as text completion (Gong et al., 2025; Nie et al., 2025; Xie et al., 2025b; Song et al., 2025). Such a masked modeling paradigm that distinguishes masked tokens from unmasked tokens allows MDMs to offer the efficiency of parallel decoding while maintaining high-quality performance.

Despite the advantages, MDMs inherently suffer from factorization errors when sampling multiple tokens simultaneously, which necessitates costly iterative decoding in practice (Deschenaux & Gulcehre, 2025; Xu et al., 2025; Liu et al., 2025; Hayakawa et al., 2025; Yoo et al., 2025). To address the computational burden associated with such iterative processes, various few-step distillation methods (Deschenaux & Gulcehre, 2025; Hayakawa et al., 2025; Yoo et al., 2025) have been actively explored. However, even with few-step distillation techniques, challenges in few-step generation persist, suggesting the presence of an inherent performance ceiling regardless of the distillation method. Specifically, as MDM utilizes a single-masked state that absorbs the entire data distribution, the stochasticity in the data distribution must originate from categorical sampling. However, as the simultaneous decoding of multiple tokens incurs inevitable factorization error, MDMs cannot fully match the target distribution in extremely few-step regimes, even if the model is at the optimum with respect to its objective.

In this paper, we address these theoretical and practical challenges through the following contributions:

- First, we analyze the factorization error in MDMs, identifying a non-trivial theoretical lower bound that persists independently of the distillation method employed due to their use of a deterministic single-state mask.

- Second, to overcome this theoretical limit, we propose

the Infinite Mask Diffusion Model (IMDM). By replacing the deterministic mask token with stochastic latent masks that remain strictly distinguishable from valid data tokens, IMDM addresses the inherent factorization error bound while leveraging the advantages of MDMs, including their simple framework and effective conditional generation capabilities.

- Finally, we show that integrating IMDM with existing distillation frameworks serves as an effective strategy to enhance the few-step generation performance of MDMs.

## 2. Related Works

**Masked Diffusion Models**  Masked Diffusion Models (MDMs) are a class of discrete diffusion models that utilize absorbing states as priors (Austin et al., 2021; Lou et al., 2024; Sahoo et al., 2024; Shi et al., 2024). In these models, the forward process progressively masks each token in the data, and the model learns to predict the masked tokens based on the unmasked context tokens. This framework is simple and effective, leveraging a mask token that does not appear in the data, which leads to a straightforward objective and decoding process (Sahoo et al., 2024; von Rütte et al., 2025). Moreover, MDMs can adapt easily to conditional generative tasks, as their training involves generating data with arbitrarily shaped masked regions (Lou et al., 2024; Nie et al., 2025; Ou et al., 2025).

The simplicity of the framework, combined with its potential for handling various masking patterns, makes MDMs a promising alternative to autoregressive (AR) models (Nie et al., 2025), particularly for generative tasks such as text completion and code infilling. Unlike AR models (Radford et al., 2019; Grattafiori et al., 2024), which use sequential decoding with causal transformers, MDMs benefit from parallel decoding and bidirectional context processing, which improves efficiency and generation quality (Nie et al., 2025). Recent research (Bie et al., 2025; Zhu et al., 2025; Gong et al., 2025; Xie et al., 2025b) has applied MDMs to various generative downstream tasks, demonstrating their effectiveness in handling complex token dependencies through the masking process.

**Few-Step Distillation for MDMs**  Despite their effectiveness, Masked Diffusion Models (MDMs) often require many sampling steps to ensure high-quality generation, primarily due to factorization errors that arise from simultaneous token updates, which disrupt token correlations (Deschenaux & Gulcehre, 2025; Hayakawa et al., 2025; Yoo et al., 2025; Xu et al., 2025; Liu et al., 2025). Recent approaches (Deschenaux & Gulcehre, 2025; Hayakawa et al., 2025) have focused on few-step distillation to reduce the number of sampling iterations needed for high-quality gen-

eration. These methods involve distilling many-step predictions of a teacher model into few-step predictions of a student to achieve similar performance with fewer steps.

Although progress has been made, we observe that a theoretical lower bound exists for the factorization error that MDMs cannot overcome. This lower bound arises from the deterministic nature of the single-state mask (*i.e.,* the fully-masked state) used in MDMs, which prevents them from fully capturing token dependencies in fewer steps. To address the bottleneck, we propose the Infinite Mask Diffusion Model (IMDM), which introduces stochasticity to masked tokens to enable effective few-step generation.

While Di4C (Hayakawa et al., 2025) also explores stochasticity, the method employs a mixture model that results in complex objectives sensitive to hyperparameters. In contrast, IMDM reinterprets the single-state mask as an infinite mask, providing a simpler and more direct adaptation of existing distillation methods while incorporating stochasticity. We provide an extended comparison to other recent approaches that extend or analyze discrete diffusion beyond the standard MDM formulation in Sec. D.

## 3. Background

**Notation**  We consider a discrete random variable $X$ taking values in the space of one-hot vectors $\mathcal{V} = \{\mathbf{x} \in \{0,1\}^K : \sum_{i=1}^{K} x_i = 1\}$. We denote the $K$-dimensional column vector of ones by $\mathbf{1}$ and a one-hot vector that represents a category $k \in \{1, 2, \cdots, K\}$ as $\mathbf{i}_k$. We use $\odot$ and $\langle \cdot, \cdot \rangle$ for the Hadamard and dot products, respectively. For a sequence of length $L$, we denote the sequence as $\mathbf{x}^{1:L} = (\mathbf{x}^1, \ldots, \mathbf{x}^L) \in \mathcal{V}^L$, where the $\ell$-th token is given by $\mathbf{x}^\ell$. $\mathrm{Cat}(\cdot; \boldsymbol{p})$ denotes the categorical distribution over $\mathcal{V}$ parameterized by the probability vector $\boldsymbol{p} \in \Delta^{K-1}$, where $\Delta^{K-1}$ is a simplex over $K$ categories.

### 3.1. Discrete Diffusion Models

**Discrete Diffusion Models**  Building on the formulation of Discrete Denoising Diffusion Probabilistic Models (D3PMs) (Austin et al., 2021), we adopt the framework proposed by Sahoo et al. (2024), which defines the forward process as an interpolation between the clean data $\mathbf{x}$ and a noise distribution $\boldsymbol{\pi} \in \Delta^{K-1}$. Given a monotonically decreasing noise schedule $\alpha_t \in [0, 1]$, the forward transition probability $q(\mathbf{z}_t|\mathbf{x})$ is defined as:

$$q(\mathbf{z}_t|\mathbf{x}) = \mathrm{Cat}(\mathbf{z}_t; \alpha_t \mathbf{x} + (1 - \alpha_t)\boldsymbol{\pi}), \qquad (1)$$

where $\mathbf{z}_t \in \mathcal{V}$ denotes the latent variable at time step $t$. For any $s < t$, the posterior distribution $q(\mathbf{z}_s|\mathbf{z}_t, \mathbf{x})$ takes the

following closed form using the ratio $\alpha_{t|s} = \alpha_t / \alpha_s$:

$$q(\mathbf{z}_s | \mathbf{z}_t, \mathbf{x}) = \text{Cat}$$

$$\left( \mathbf{z}_s; \frac{[\alpha_{t|s}\mathbf{z}_t + (1 - \alpha_{t|s})\mathbf{1}\boldsymbol{\pi}^\top \mathbf{z}_t] \odot [\alpha_s \mathbf{x} + (1 - \alpha_s)\boldsymbol{\pi}]}{\alpha_t \mathbf{z}_t^\top \mathbf{x} + (1 - \alpha_t)\mathbf{z}_t^\top \boldsymbol{\pi}} \right). \tag{2}$$

A common choice for training the diffusion model is optimizing the standard variational lower bound (ELBO) (Austin et al., 2021; Sahoo et al., 2025; Schiff et al., 2025; Sahoo et al., 2024; Shi et al., 2024). In general, the reverse process $p_\theta(\mathbf{z}_s | \mathbf{z}_t)$ is trained to match the posterior $q(\mathbf{z}_s | \mathbf{z}_t, \mathbf{x})$ by minimizing the Kullback-Leibler (KL) divergence:

$$\mathcal{L} = \mathbb{E}_{\mathbf{z}_s, \mathbf{z}_t} [w_{s,t} D_{KL}[q(\mathbf{z}_s | \mathbf{z}_t, \mathbf{x}) || p_\theta(\mathbf{z}_s | \mathbf{z}_t)]], \tag{3}$$

where $w_{s,t}$ weights the KL divergence. The models often parameterize the reverse transition kernel as $p_\theta(\mathbf{z}_s | \mathbf{z}_t) = q(\mathbf{z}_s | \mathbf{z}_t, \mathbf{x} = \mathbf{x}_\theta(\mathbf{z}_t, t))$, and generate data with the learned reverse process.

This framework generalizes various discrete diffusion processes depending on the choice of the prior $\boldsymbol{\pi}$ (Austin et al., 2021; Lou et al., 2024; Ou et al., 2025; von Rütte et al., 2025). Typically, $\boldsymbol{\pi}$ is set to either a uniform distribution (Sahoo et al., 2025; Schiff et al., 2025) or an absorbing state, the latter of which corresponds to Masked Diffusion Models (MDMs) (Sahoo et al., 2024; Shi et al., 2024; Nie et al., 2025). Specifically, uniform discrete diffusion (Schiff et al., 2025; Sahoo et al., 2025) defines the prior $\boldsymbol{\pi} = \frac{1}{K}\mathbf{1}$, and MDMs define the prior as $\boldsymbol{\pi} = \mathbf{m}$ while extending the state space to $\mathcal{V}^+ = \mathcal{V} \cup \{\mathbf{m}\}$, where $\mathbf{m}$ denotes a one-hot vector corresponding to the mask token.

**Advantages of Distinguishable Mask Tokens** The key property of the mask token $\mathbf{m}$ is the explicit distinction between $\mathbf{m}$ and data $\mathbf{x}$, *i.e.,* $\langle \mathbf{m}, \mathbf{x} \rangle = 0$. This disjoint property provides a simple training and inference formulation of MDMs. For instance, Sahoo et al. (2024) have shown that the posterior in Eq. (2) simplifies to:

$$q(\mathbf{z}_s | \mathbf{z}_t, \mathbf{x}) = \begin{cases} \text{Cat}(\mathbf{z}_s; \mathbf{z}_t) & \mathbf{z}_t \neq \mathbf{m}, \\ \text{Cat}(\mathbf{z}_s; \frac{(1 - \alpha_s)\mathbf{m} + (\alpha_s - \alpha_t)\mathbf{x}}{1 - \alpha_t}) & \mathbf{z}_t = \mathbf{m}, \end{cases} \tag{4}$$

providing a simple decoding process that progressively unmasks the masked tokens. Moreover, with Rao-Blackwellization, the loss term in Eq. (3) simplifies to:

$$\mathcal{L} = \mathbb{E}_{\mathbf{z}_t, t} \left[ \frac{\alpha'_t}{1 - \alpha_t} \log(\langle \mathbf{x}_\theta(\mathbf{z}_t, t), \mathbf{x} \rangle) \right], \tag{5}$$

where $\alpha'_t = \frac{\partial \alpha_t}{\partial t}$.

MDM's use of a distinguishable mask token is particularly beneficial for conditional generation. By explicitly indicating which positions are observed (conditioning context) and

which must be predicted, the mask token enables a clean separation between context encoding and target decoding. As a result, diverse conditioning patterns (including prefix completion and span infilling) can be implemented simply by selecting the mask region, leading to robust performance on conditional generation tasks (*e.g.,* text completion) in both finetuned and training-free settings (Gong et al., 2025; Nie et al., 2025; Xie et al., 2025b; Shi et al., 2024).

### 3.2. Factorization Error of Discrete Diffusion Models

Discrete diffusion models typically avoid modeling the full joint transition kernel $p_\theta(\mathbf{z}_t | \mathbf{z}_s)$, since doing so would require predicting a categorical distribution over $K^L$ joint configurations, *i.e.,* a probability vector in $\Delta^{K^L - 1}$, which is computationally infeasible for most sequences. Instead, they use the factorized approximation $p_\theta(\mathbf{z}_t^{1:L} | \mathbf{z}_s) \approx \prod_{\ell=1}^{L} p_\theta(\mathbf{z}_t^\ell | \mathbf{z}_s)$, which scales to high-dimensional data but discards dependencies among tokens. We refer to the resulting mismatch between the joint and the model's factorized prediction as *factorization error*, which can be quantified via the model-dependent conditional total correlation (Yoo et al., 2025; Hayakawa et al., 2025; Liu et al., 2025):

$$TC_\theta(\mathbf{z}_s | \mathbf{z}_t) = \mathbb{E}_{\mathbf{z}_t} \left[ D_{KL}(p(\mathbf{z}_s | \mathbf{z}_t) || \prod_{\ell=1}^{L} p_\theta(\mathbf{z}_s^\ell | \mathbf{z}_t)) \right]. \tag{6}$$

$TC_\theta$ measures the gap between the true joint and the model's factorized prediction, and reduces to the data-only total correlation when $p_\theta$ matches the true per-token marginals. Specifically, since the factorized model treats the posterior of each token independently, simultaneous updates to jointly correlated tokens distort the true distribution, resulting in poor generation quality. In few-step generation, larger time gaps between $s$ and $t$ necessitate simultaneous updates of multiple tokens, making the addressing of factorization error essential for high-quality generation (Yoo et al., 2025; Hayakawa et al., 2025). To mitigate the error, prior works aim to reduce the discrepancy by approximating the joint posterior with a teacher model's multi-step prediction (Deschenaux & Gulcehre, 2025; Hayakawa et al., 2025) or with a rectified coupling constructed by generating data with a teacher model (Yoo et al., 2025).

## 4. Method

As discussed in Sec. 3.1, the advantages of MDMs appear to stem primarily from the clear distinction between masked and valid data tokens (*i.e.,* $\langle \mathbf{x}, \mathbf{m} \rangle = 0$). This observation raises the question of whether a single, deterministic mask token is strictly necessary to maintain these benefits.

In this section, we first argue that the use of a single deterministic mask token results in the irreducible factorization error that acts as a fundamental bottleneck for existing few-

step distillation methods (Sec. 4.1). Then, we propose the Infinite Mask Diffusion Model (IMDM), which introduces infinite stochastic mask tokens to MDMs (Sec. 4.2). By leveraging the distinction property of MDM while avoiding the use of a single, deterministic mask token, IMDM mitigates the theoretical lower bound of the factorization error, while leveraging MDM's simple framework and effective conditional generation.

### 4.1. Factorization Error Bound of MDMs

As discussed in Sec. 3.2, the factorization error arises when correlated tokens are updated simultaneously. In this section, we argue that the use of a single, deterministic mask token incurs an irreducible factorization error lower bound. To theoretically analyze the lower bound of this error specifically in MDMs, we consider a scenario where the $i$-th token $\mathbf{z}_t^i$ and the $j$-th token $\mathbf{z}_t^j$ are both masked at timestep $t$ and unmasked between $t$ and $s$. Let $p(e_{ij})$ denote the probability of this specific event occurring.

**Theorem 4.1** (Lower Bound of Factorization Error in MDMs). *Let $e_{ij}$ be the event where tokens at indices $i$ and $j$ are simultaneously decoded. Then, for any MDM with parameters $\theta$, the conditional total correlation is lower-bounded by their conditional mutual information weighted by the event probability:*

$$TC_\theta(\mathbf{z}_s|\mathbf{z}_t) \geq \max_{i,j} p(e_{ij}) \cdot I(\mathbf{z}_s^i; \mathbf{z}_s^j \mid \mathbf{z}_t^{\backslash\{i,j\}}), \quad (7)$$

*where $\mathbf{z}_t^{\backslash\{i,j\}}$ denotes the set of tokens at timestep $t$ excluding the tokens at indices $i$ and $j$.*

*Sketch of Proof.* Since $TC_\theta(\mathbf{z}_s|\mathbf{z}_t) \geq TC(\mathbf{z}_s|\mathbf{z}_t)$ for any $\theta$, where $TC(\mathbf{z}_s|\mathbf{z}_t) = \mathbb{E}_{\mathbf{z}_t}[D_{KL}(p(\mathbf{z}_s|\mathbf{z}_t)\| \prod_\ell p(\mathbf{z}_s^\ell|\mathbf{z}_t))]$ is the data-only total correlation, it suffices to lower-bound $TC$. We derive the following bound by leveraging the distinction property of MDM (*i.e.,* $\langle \mathbf{x}, \mathbf{m} \rangle = 0$):

$$TC(\mathbf{z}_s|\mathbf{z}_t) \geq \mathrm{LB}(e_{ij}), \text{ where}$$
$$\mathrm{LB}(e_{ij}) = p(e_{ij})\bigg[I(\mathbf{z}_s^i; \mathbf{z}_s^j \mid \mathbf{z}_t^{\backslash\{i,j\}}, e_{ij})$$
$$- H\left((\mathbf{z}_t^i, \mathbf{z}_t^j) \mid \mathbf{z}_t^{\backslash\{i,j\}}, e_{ij}\right)\bigg]. \quad (8)$$

Then, we use the following equation as the states of masked tokens are deterministic (*i.e.,* $H(\mathbf{z}_i) = H(\mathbf{z}_j) = 0$):

$$H\left((\mathbf{z}_t^i, \mathbf{z}_t^j) \mid \mathbf{z}_t^{\backslash\{i,j\}}, e_{ij}\right) = 0. \quad (9)$$

Substituting these into Eq. (8), and noting that the inequality holds for any pair of indices $i$ and $j$, we obtain the stated lower bound by taking the maximum over all pairs. The complete proof is provided in Sec. A.1. □

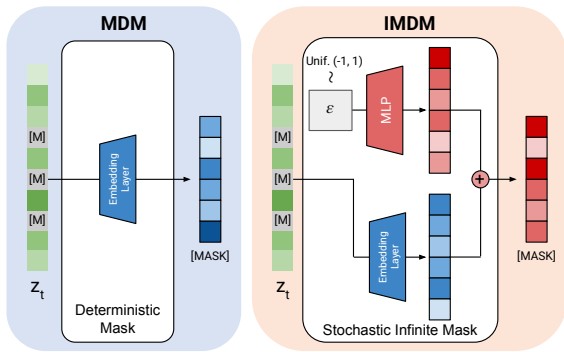

*Figure 1.* An overview figure of IMDM. In IMDM, $\epsilon$ is sampled from a uniform distribution and processed through an MLP to generate a stochastic component, which is then added to the base mask embedding. This addition renders the final mask embedding stochastic, inheriting the randomness of $\epsilon$, and enabling the model to simulate an infinite variety of masks.

The theorem suggests that the lower bound of the factorization error becomes more pronounced in two primary scenarios. First, the error bound tends to increase as the step size grows, a characteristic of few-step generation. As indicated in Eq. (4), the first term in the bound, $p(e_{ij})$, scales with $(\frac{\alpha_s-\alpha_t}{1-\alpha_t})^2$. This relationship implies that the lower bound rises as the step size $\Delta_t = (\alpha_t - \alpha_s)$ increases, particularly as the timestep $t$ approaches the data distribution ($\alpha_t \to 1$). Consequently, the factorization error lower bound of MDMs may pose a significant challenge in few-step decoding regimes, which necessitate larger step sizes.

The second scenario involves a significant value of the conditional mutual information term $I(\mathbf{z}_s^i; \mathbf{z}_s^j \mid \mathbf{z}_t^{\backslash\{i,j\}}, e_{ij})$, which quantifies the correlation between unmasked tokens $\mathbf{z}_s^i$ and $\mathbf{z}_s^j$ given the previous state $\mathbf{z}_t^{\backslash\{i,j\}}$. Given that discrete diffusion models are widely applied to natural language tasks which usually have a strong dependency between tokens, it is reasonable to assume the existence of token pairs exhibiting non-negligible mutual information. These observations collectively suggest that the presence of this theoretical bound may constrain the ability of MDMs to generate semantically consistent data within the few-step regime. Furthermore, as this lower bound is inherent to the formulation and independent of the specific model architecture, it likely limits the potential gains from distillation, even when employing powerful optimization techniques.

### 4.2. Infinite Mask Diffusion Model

In Sec. 4.1, we observed that the lower bound of factorization error, which is inherent to the MDM framework, poses a potential barrier to few-step generation, even when distillation techniques are applied.

To address the inherent limitations of MDMs while preserving their simple design and effective generation performance, we propose the Infinite Mask Diffusion Model (IMDM)

(IMDM), as illustrated in Fig. 1. Motivated by the observation that the benefits of MDMs likely stem from the distinguishability of the mask token rather than its deterministic single-state nature, we formulate IMDM to utilize an infinite set of stochastic mask tokens that remain strictly distinguishable from the data. Specifically, we start the derivation of IMDM from uniform discrete diffusion over an extended state space and show that it reduces to a purely masked diffusion form by enforcing the distinguishability between data and mask tokens. This derivation strategy allows IMDM to directly inherit the NELBO of uniform discrete diffusion (Schiff et al., 2025) (see Sec. A.3).

In the following paragraphs, we first introduce the formulation of IMDM, highlighting that its training and decoding processes closely mirror the simple structure of standard MDMs. Next, we provide a theoretical analysis suggesting that the factorization error lower bound observed in MDMs is effectively mitigated in our framework; specifically, we demonstrate the existence of an optimal model capable of minimizing this error. Finally, we discuss the compatibility between IMDM and MDM, providing implementation details that allow for the efficient reuse of pre-trained MDM weights.

**IMDM Formulation**   To simultaneously achieve the distinguishability of mask tokens and the flexibility of infinite stochastic masks, we define the state space $\mathcal{Z}$ as the union of the data vocabulary $\mathcal{V}$ and an infinite set of latent mask states $\mathcal{M}$ (where $|\mathcal{M}| \to \infty$). Recognizing that the explicit separation between masked and unmasked tokens facilitates the simple formulation and effective conditional generation of MDMs, we enforce a strict disjoint property between the data and the set of latent mask states as follows:

$$\mathcal{Z} = \mathcal{V} \cup \mathcal{M}, \quad \text{where } \mathcal{V} \cap \mathcal{M} = \emptyset. \tag{10}$$

To introduce stochasticity to the mask tokens, we formulate the IMDM forward process as uniform discrete diffusion over the extended state space $\mathcal{Z}$. Specifically, the forward transition probability of IMDM $q_{\text{IMDM}}(\mathbf{z}_t|\mathbf{x})$ is defined as:

$$q_{IMDM}(\mathbf{z}_t|\mathbf{x}) = \alpha_t \mathbf{x} + \frac{1 - \alpha_t}{K} \mathbf{1}$$
$$= \alpha_t \mathbf{x} + (1 - \alpha_t) \left( \frac{N}{K} \boldsymbol{\pi}_{\mathcal{V}} + \frac{M}{K} \boldsymbol{\pi}_{\mathcal{M}} \right), \tag{11}$$

where $N = |\mathcal{V}|$ and $M = |\mathcal{M}|$ denote the sizes of the data and latent mask categories, respectively, and $K = N + M$ is the total dimension. The vectors $\boldsymbol{\pi}_{\mathcal{V}}, \boldsymbol{\pi}_{\mathcal{M}}$ denote probability vectors representing a uniform distribution over $\mathcal{V}$ and $\mathcal{M}$ respectively. Considering the limit $M \to \infty$, the terms simplify (since $N/K \to 0$ and $M/K \to 1$), yielding:

$$q_{IMDM}(\mathbf{z}_t|\mathbf{x}) = \alpha_t \mathbf{x} + (1 - \alpha_t) \boldsymbol{\pi}_{\mathcal{M}}. \tag{12}$$

The resulting forward process closely resembles MDM while a token transitions to one of the latent mask states in $\mathcal{M}$ uniformly at random with probability $1 - \alpha_t$. Furthermore, similar to Sahoo et al. (2024), which relies on the separation between the mask token and data, IMDM preserves the distinction property (*i.e.,* $\langle \mathbf{z}, \mathbf{x} \rangle = 0$ for all $\mathbf{z} \in \mathcal{M}$). Leveraging the property, the posterior distribution is given by:

$$q(\mathbf{z}_s|\mathbf{z}_t, \mathbf{x}) = \begin{cases} \text{Cat}(\mathbf{z}_s; \mathbf{z}_t) & \mathbf{z}_t \in \mathcal{V}, \\ \text{Cat}(\mathbf{z}_s; \frac{(1-\alpha_s)\boldsymbol{\pi}'_{\mathcal{M}} + (\alpha_s - \alpha_t)\mathbf{x}}{1-\alpha_t}) & \mathbf{z}_t \in \mathcal{M}, \end{cases} \tag{13}$$

where $\boldsymbol{\pi}'_{\mathcal{M}} = \alpha_{t|s} \mathbf{z}_t + (1 - \alpha_{t|s}) \boldsymbol{\pi}_{\mathcal{M}}$.

In addition to the posterior distribution, the training objective of IMDM is identical to MDM's objective in Eq. (3):

$$\mathcal{L}_{IMDM} = \mathbb{E}_{\mathbf{z}_t, t} \left[ \frac{\alpha'_t}{1 - \alpha_t} \log(\langle \mathbf{x}_\theta(\mathbf{z}_t, t), \mathbf{x} \rangle) \right], \tag{14}$$

where $\alpha'_t = \frac{\partial \alpha_t}{\partial t}$. Eq. (14) shares the same form as MDM's NELBO, demonstrating the seamless design of IMDM, and is used only to verify that pre-trained MDM weights transfer to IMDM (Sec. C.4); our few-step generation experiments instead train with distillation methods (see Sec. 5.3). Full derivations regarding the posterior and NELBO can be found in Sec. A.3.

As shown in Eqs. (12) and (14), IMDM closely resembles the simple and effective framework of MDMs. By resembling the simple structure with explicit distinction between masked and unmasked tokens, IMDM can inherit the benefits of MDMs, including their performance on conditional generation. More precisely, identifying $\mathcal{M}$ as the single token $\mathbf{m}$ collapses the IMDM state space $\mathcal{Z} = \mathcal{V} \cup \mathcal{M}$ to the MDM state space $\mathcal{V}^+ = \mathcal{V} \cup \{\mathbf{m}\}$ and reduces the IMDM forward of Eq. (12) to the standard MDM forward. This structural relationship enables seamless weight transfer from a pretrained MDM to IMDM, which we further confirm empirically in Sec. C.4. The next subsection shows that keeping $\mathcal{M}$ uncollapsed admits an optimal model that mitigates the factorization-error bound of Theorem 4.1.

**Implementation of IMDM**   Although the theoretical framework closely mirrors that of MDMs, the practical success of IMDM in few-step distillation depends on the ability to effectively utilize pre-trained MDM parameters. In order to seamlessly adapt pre-trained MDMs for IMDM distillation, we define the embedding layer $E_\theta : \mathcal{Z} \to \mathbb{R}^d$ as:

$$E_\theta(\mathbf{z}) = \begin{cases} E_\psi(\mathbf{z}) & \mathbf{z} \in \mathcal{V}, \\ E_\psi(\mathbf{m}) + \text{MLP}_\theta(\epsilon) & \mathbf{z} \in \mathcal{M}, \end{cases} \tag{15}$$

where $\psi$ denotes the parameters of the pre-trained MDM, $\mathbf{m}$ represents the mask token of the pre-trained model, and $\epsilon \in \mathbb{R}^d$ is a continuous noise vector that follows

Unif.$(-1, 1)$. Since explicitly instantiating the infinite set $\mathcal{M}$ is intractable, this formulation allows us to represent the mask states implicitly by injecting sampled continuous noise $\epsilon$ into the fixed mask embedding. We initialize the weights and biases of the output layer of $\mathrm{MLP}_\theta$ to zero, ensuring that the IMDM initially behaves identically to the pre-trained MDM. This design effectively models a high-entropy mask state (approximating $|\mathcal{M}| \to \infty$) while preserving the observable behavior of the original MDM, thereby enabling the seamless reuse of pre-trained weights. Consequently, the compatible design allows for the direct application of existing distillation methods on the IMDM framework.

**Distillation with IMDM**  The main goal of IMDM is to mitigate the theoretical limitation of MDMs on few-step generation. By introducing stochasticity, IMDM guarantees the existence of parameter $\theta$ that reduces the factorization error to 0.

**Theorem 4.2** (Zero Factorization Error in IMDM). *There exist an IMDM forward process and parameters $\theta^*$ such that*

$$TC_{\theta^*}(\mathbf{z}_s | \mathbf{z}_t) = 0 \quad \text{for all } \mathbf{z}_s, \mathbf{z}_t \in \mathcal{Z}. \tag{16}$$

*Proof.* The proof is provided in Sec. A.2.  $\square$

The theorem suggests that, unlike MDMs which appear to be limited by a structural lower bound of factorization error, IMDM offers a theoretical capability to mitigate this error when the factorization error captured by Eq. (16) is reduced.

Concretely, IMDM admits a construction of an optimal model via a partition-and-map mechanism that exploits the stochasticity of its mask states. Let $E$ denote the set of positions that are masked at $t$ and unmasked between $t$ and $s$. Conditioned on the unmasked context $\mathbf{z}_t^{E^C}$, the construction defines a deterministic map $F : \mathcal{M}^{|E|} \to \mathcal{V}^{|E|}$ that takes the mask-state values $\mathbf{z}_t^E$ as input and returns a token sequence in $\mathcal{V}^{|E|}$. Since $|\mathcal{M}| \to \infty$, $\mathbf{z}_t^E$ admits arbitrarily fine partitions whose subset probabilities match those of $p(\mathbf{z}_s^E | \mathbf{z}_t)$, and $F$ deterministically assigns each subset to a valid sequence; marginalizing over $\mathbf{z}_t^E$ then recovers the true conditional. In contrast, MDMs use a single deterministic mask token, so $\mathbf{z}_t^E$ takes a single fixed value with no variation to partition; the MDM model is therefore restricted to factorized predictions and incurs the structural lower bound formalized in Theorem 4.1. We emphasize that this construction shows the existence of an optimal IMDM, not that the trained model automatically realizes it. The formal construction is provided in Sec. A.2, and we empirically validate this prediction in Sec. 5.2.

We interpret the iterative distillation procedures of SDTT (Deschenaux & Gulcehre, 2025), ReDi (Yoo et al., 2025), and Di4C (Hayakawa et al., 2025) as progressively

reducing the factorization error of Eq. (16): SDTT iteratively transfers longer effective rollouts into 1-step student predictions; ReDi iteratively rectifies the coupling between $\mathbf{z}_t$ and $\mathbf{x}$, an empirical analog of the construction in Theorem 4.2's proof; Di4C distills dimensional correlations across rounds.

# 5. Experiments

In this section, we evaluate our proposed method, IMDM, across three distinct scenarios. We first introduce the experimental setup for standard language modeling benchmarks (Sec. 5.1). Next, we validate the theoretical implication of the factorization error bound and analyze the role of infinite mask of IMDM using a controlled synthetic dataset (Sec. 5.2). Then, we conduct an in-depth analysis of distillation strategies and model compatibility on the One Billion Word (LM1B) dataset (Sec. 5.3). Finally, we demonstrate the few-step generation performance of IMDM on OpenWebText benchmark (Sec. 5.4).

## 5.1. Experimental Setup for Language Modeling

In this subsection, we detail the configuration shared across our language modeling experiments on LM1B and OpenWebText. Specific details for the synthetic dataset are provided in Sec. 5.2.

**Datasets and Models**  We utilize two standard benchmarks: One Billion Word (LM1B) (Chelba et al., 2013) and OpenWebText (Gokaslan & Cohen, 2019). Following Sahoo et al. (2025), we set the sequence length to 128 tokens for LM1B and 1024 tokens for OpenWebText, employing sentence-packing (Raffel et al., 2020) to match these lengths. For tokenization, we use the BERT tokenizer (Devlin et al., 2019) (vocab size 30,522) for LM1B and the GPT-2 tokenizer (Radford et al., 2019) (vocab size 50,257) for OpenWebText. Consistent with prior works (Sahoo et al., 2024; Deschenaux & Gulcehre, 2025), we adopt the GPT-2 Small architecture for all experiments. Further implementation details are provided in Sec. B.

**Evaluation Metrics**  We evaluate generation quality primarily using Generative Perplexity and Sample Entropy (Dieleman et al., 2022). Following standard protocols (Hayakawa et al., 2025; Sahoo et al., 2025), we generate 1,024 samples and compute generative perplexity using a pre-trained GPT-2 Large model (Radford et al., 2019). To complement generative perplexity and identify potential failure modes such as mode collapse, we additionally report Sample Entropy (Wang et al., 2022). For the conditional generation experiment on OpenWebText, we further employ the MAUVE score (Pillutla et al., 2021) to assess the alignment between the generated continuations and the given

*Table 1.* Synthetic experiment result. Validity denotes the ratio of valid samples in generated samples, and token entropy denotes the entropy of marginal distribution. 'Error' denotes the factorization error in Eq. (6) measured at full masking by marginalizing over 10,000 noise realizations.

| Models | Validity | Token Entropy | Error |
|---|---|---|---|
| MDM | 49.8% | 0.69 | 0.693 |
| IMDM | 97.7% | 0.69 | 0.082 |
| Random Seq. | 50.0 % | 0.69 | – |

*Table 2.* Per-token output probabilities $P(\text{token} = 0)$ on the synthetic $\{00, 11\}$ task. IMDM yields near-deterministic predictions that shift coherently with the noise realization $\epsilon$, while MDM yields near-uniform predictions regardless of input.

| Model | Noise | Token 1 | Token 2 |
|---|---|---|---|
| IMDM | $\epsilon_A$ | 98.2% | 100.0% |
| IMDM | $\epsilon_B$ | 0.6% | 0.4% |
| MDM | – | 49.7% | 49.7% |

prompts (Deschenaux & Gulcehre, 2025). In this conditional setting, following Deschenaux & Gulcehre (2025), we randomly select 256 prompts (consisting of the first 50 tokens) from the test set and generate five completions of 1,024 tokens per prompt, using the first 100 generated tokens for evaluation. All reported results use standard categorical sampling without temperature, top-k, or top-p adjustments.

### 5.2. Empirical Analysis on Synthetic Dataset

We first demonstrate the practical implications of the factorization error bound discussed in Sec. 4.1 using a synthetic dataset, which MDMs inherently fail to generate in a single step due to the deterministic nature of the mask token. The synthetic dataset consists of binary sequences $\{00, 11\}$, where a sequence is considered valid only if the constituent tokens are identical. Due to the perfect correlation between the tokens, standard MDMs are theoretically incapable of generating valid data with single-step decoding. Specifically, since the model predicts each token independently from the single mask token, it samples each token from an independent uniform distribution (50:50), failing to capture the dependency.

To empirically validate that MDM has non-reducible error, we first train a standard MDM on the dataset and subsequently perform distillation using ReDi (Yoo et al., 2025) to obtain both a distilled MDM and a distilled IMDM. We evaluated the models by generating 5,000 samples with one-step decoding and measuring two key metrics. First, we measured Validity, defined as the ratio of generated sequences where the tokens are identical. Second, we measured Token-Entropy, which evaluates the diversity of characters across the entire generated dataset (Ground Truth = $\log 2 \approx 0.69$). Closer alignment with the Ground Truth (GT) values indicates better performance.

Tab. 1 presents the 1-step generation performance of each model. While both MDM and IMDM accurately match the token-level marginal distribution, MDM fails to capture the inter-token correlations, resulting in a validity score comparable to that of random sampling. In contrast, IMDM achieves near-perfect validity, demonstrating its capability to learn the underlying data structure and generate consistent sequences aligned with the Ground Truth

(GT). The factorization-error column further sharpens this contrast: MDM saturates at the predicted lower bound of $\ln 2 \approx 0.693$ from Theorem 4.1, while IMDM falls well below it to $0.082$, confirming that the bound is structural for MDM and that IMDM lifts it as established in Theorem 4.2.

To verify that IMDM realizes the partition-and-map mechanism in Sec. 4.2, we examine the per-token output probabilities of IMDM under two distinct noise realizations $\epsilon_A$ and $\epsilon_B$. The per-token output probabilities are summarized in Tab. 2. IMDM produces near-deterministic, internally consistent predictions that nevertheless vary with the noise realization: both tokens are predicted as 0 under $\epsilon_A$, and 1 under $\epsilon_B$. This routing of distinct noise realizations to distinct coherent outcomes mirrors the partition-and-map construction underlying Theorem 4.2. In contrast, MDM outputs near-uniform marginals at both positions regardless of input, consistent with the structural barrier of Theorem 4.1.

### 5.3. Investigating IMDM on LM1B Dataset

To efficiently analyze and screen effective distillation strategies on language modeling, we conduct our design analysis on LM1B (Chelba et al., 2013). As IMDM is structurally compatible with pre-trained MDMs, all MDM-based distillation strategies are directly applicable to IMDM.

**Distillation of IMDM** While IMDM theoretically resolves the factorization error bound inherent to MDMs, its empirical advantages become most pronounced when coupled with effective distillation strategies. To validate the empirical effectiveness of IMDM, we apply IMDM to two representative distillation methods, SDTT (Deschenaux & Gulcehre, 2025) and ReDi (Yoo et al., 2025), as well as their combination (SDTT+ReDi).

For SDTT, we train the student model to match the teacher's multi-step predictions sharing the same initial state $\mathbf{z}_t$ including the continuous noise for IMDM. Following Deschenaux & Gulcehre (2025), we perform 7 rounds of distillation, with each round consisting of 10,000 iterations. For ReDi, we adopt the perturbed rectification strategy (Yoo et al., 2025) to construct a rectified coupling using data generated by the teacher. The student model is then trained using the loss form of Eq. (14), with the data distribution replaced by this rectified coupling. We utilize 100,000 pairs

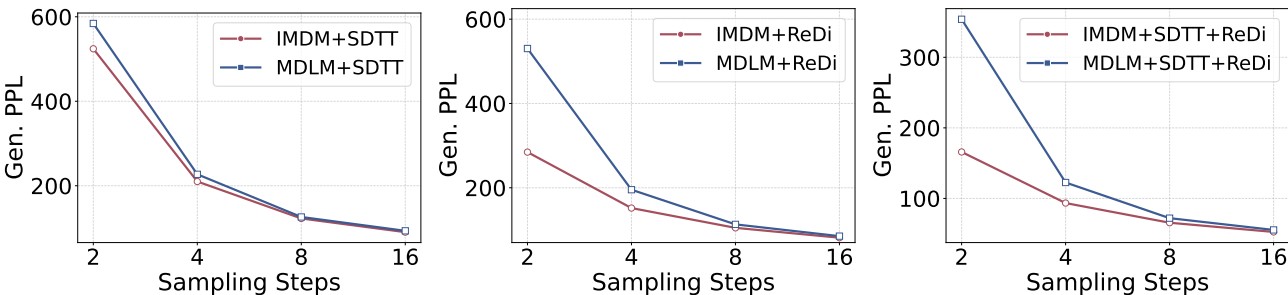

*Figure 2.* Unconditional generation on LM1B. Each figure provides a comparison between MDLM and IMDM in various distillation methods. Lower generative perplexity (Gen. PPL) indicates more natural texts. We provide the exact values in Sec. C.1.

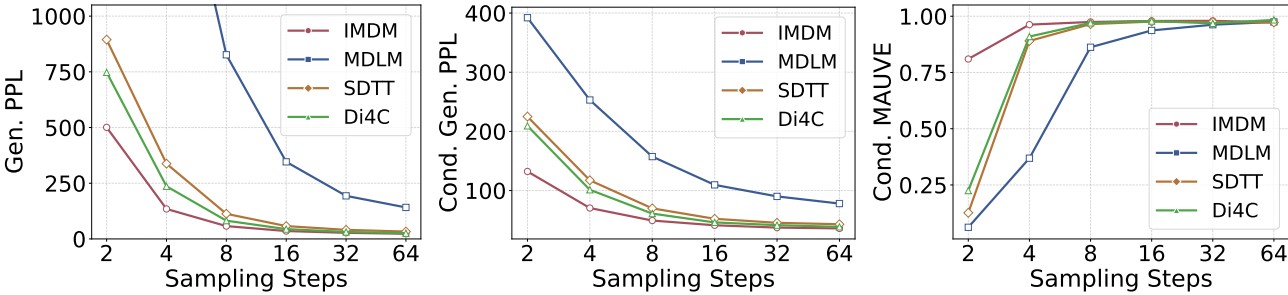

*Figure 3.* Experiment results on OpenWebText. The left plot shows the unconditional generative perplexity over decoding steps. The middle and right panels show the conditional generative perplexity and MAUVE scores over decoding steps respectively. Lower generative perplexity and higher MAUVE value indicate better generation performance. With appropriate distillation methods, IMDM surpasses the distillation baselines on both unconditional and conditional generation in the few-step regime ($\leq 8$ steps), and remains competitive at larger step counts. We provide the exact values in Sec. C.1.

for coupling construction and train for 50,000 iterations. In the SDTT+ReDi setting, we apply ReDi to a student model pre-distilled via SDTT.

The unconditional generation results on LM1B are summarized in Fig. 2. We observe that applying IMDM consistently improves performance across all distillation methods in the few-step regime ($\leq 4$ steps). Notably, the performance gain is exceptionally large in ReDi-based methods compared to SDTT alone. For instance, in the 2-step generation with ReDi, IMDM reduces the PPL by nearly 50% compared to MDLM. The gap narrows as the step count grows, which is consistent with the prediction of Theorem 4.1 (Sec. 4.1) that the factorization-error floor scales with the step size and vanishes in the many-step limit. We therefore position IMDM as a targeted remedy for the few-step bottleneck.

### 5.4. Experiments on OpenWebText Dataset

To evaluate the scalability and effectiveness of our method on large-scale benchmarks, we demonstrate IMDM on the OpenWebText dataset, covering both unconditional and conditional generation tasks. Based on the observations in Sec. 5.3, we adopt the SDTT+ReDi variant of IMDM for comparison against strong baselines, including MDLM (Teacher), SDTT, and Di4C. We utilize the official checkpoints for all baselines (Sahoo et al., 2024; Deschenaux &

Gulcehre, 2025; Hayakawa et al., 2025), specifically, the SDTT model distilled for 7 rounds and the Di4C model distilled for 2 rounds initialized from the 7-round SDTT checkpoint. Results on 860M-sized models are in Sec. C.2.

**Unconditional Generation** The unconditional generation results are presented in Fig. 3 (left). IMDM outperforms both the teacher model (MDLM) and existing distillation baselines (SDTT, Di4C) in the few-step regime ($\leq 8$ steps). Consistent with our findings in Sec. 5.3, the performance gap between IMDM and the baselines widens as the number of steps decreases. This trend supports our hypothesis that IMDM effectively mitigates the irreducible factorization error bound of MDMs, which acts as a critical bottleneck in extreme few-step generation.

**Conditional Generation** The quantitative results for conditional generation are summarized in Fig. 3 (middle, right). IMDM is particularly effective in the few-step regime ($\leq 8$ steps), achieving the lowest conditional Gen. PPL and the highest MAUVE among all evaluated methods. At larger step counts, the distillation baselines and IMDM converge to within a narrow band on both metrics, consistent with our analysis in Sec. 4.1 that the factorization-error bound that IMDM is designed to remove vanishes as the step size shrinks. Regarding the MAUVE scores (Fig. 3, right), we

*Table 3.* Ablation study on the design of injected noise (Gen.PPL on LM1B with SDTT+ReDi). Each block varies one axis while holding the other two fixed at their defaults (Uniform / $d = 768$ / scale 1.0, marked in **bold** in the header). See Tab. 8 for the sample entropy.

| Steps | Distribution | | Dimension | | | Scale | | |
|---|---|---|---|---|---|---|---|---|
| | **Uniform** | Gaussian | 256 | **768** | 1536 | 0.5 | **1.0** | 2.0 |
| **2** | 165.85 | 167.48 | 294.41 | 165.85 | 163.83 | 179.19 | 165.85 | 164.23 |
| **4** | 93.44 | 95.82 | 116.00 | 93.44 | 96.47 | 94.82 | 93.44 | 94.60 |
| **8** | 65.55 | 67.98 | 68.74 | 65.55 | 69.26 | 65.87 | 65.55 | 66.10 |
| **16** | 52.51 | 56.26 | 54.86 | 52.51 | 57.48 | 52.99 | 52.51 | 53.88 |

observe that the score of IMDM converges efficiently; at 4 sampling steps, it achieves a score comparable to that of the teacher model utilizing 64 steps. In contrast, the teacher model fails to generate meaningful text at 2 steps (MAUVE $\approx 0$), whereas IMDM maintains a high quality (MAUVE $> 0.75$). This indicates that IMDM not only inherits the conditional generation capabilities of MDMs but also successfully distills the MDM into a few-step generative model.

To further assess generation quality and diversity, we provide qualitative samples in Fig. 4. As shown in the figure, IMDM produces coherent, grammatically correct continuations given a prefix prompt. In particular, given the same crowdfunding prefix, the three samples diverge into distinct on-topic uses of the stated $50,000 funding goal (propulsion-system development, control-system design, and imaging hardware), rather than collapsing onto a single completion. Overall, these examples illustrate that IMDM produces diverse, on-topic conditional generation in the highly compressed few-step regime where the teacher MDM struggles to generate meaningful text.

### 5.5. Ablation on Noise Design

To examine the sensitivity of IMDM to the design of the injected noise, we conduct an ablation study on three axes of the noise specification. We use the LM1B dataset with the SDTT+ReDi distillation setup of Sec. 5.3. We vary the noise distribution, the noise dimensionality, and the noise scale, and report the resulting Gen. PPL across decoding step counts in Tab. 3. We also compare IMDM against its counterpart using a finite number of mask types in Sec. C.3.

As shown in Tab. 3, IMDM is robust to the choice of noise distribution and scale, with Gen. PPL changing only marginally across these settings. In contrast, noise dimensionality plays an important role: a dimension of 768 yields strong performance, while reducing it to 256 leads to a notable increase in 2-step Gen. PPL from 165.85 to 294.41. This pattern supports the interpretation of the injected noise as a mask category simulator, where the key requirement is sufficient distinctness across mask embeddings, a condition that is readily satisfied by an appropriate choice of dimension. We additionally provide an OpenWebText counterpart of this dimension ablation in Sec. C.5, where we find that $d_{\text{noise}} = 2048$ is sufficient.

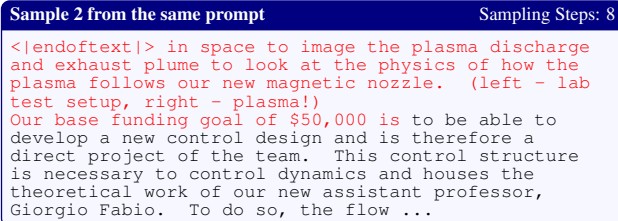

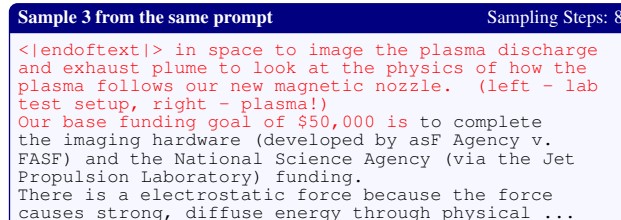

*Figure 4.* 8-step conditional generation results on OpenWebText dataset. We present the generated samples from IMDM when the prompt (red) is given. IMDM demonstrates feasible and diverse conditional generation results while fulfilling the context.

## 6. Conclusion

In this paper, we introduce the Infinite Mask Diffusion Model (IMDM), which addresses the structural factorization-error lower bound that Masked Diffusion Models (MDMs) inherit from their deterministic single-state mask by incorporating a stochastic infinite-state mask, freeing IMDM from this bound while retaining the simplicity and effectiveness of standard MDMs and enabling more effective capture of token dependencies in fewer sampling steps. We support this with both theoretical evidence showing that MDMs are bounded away from zero factorization error in few-step generation while IMDM is not, as well as empirical evidence consistent with this structural difference, as IMDM combined with appropriate distillation methods surpasses existing few-step distillation techniques on benchmarks like LM1B and OpenWebText at the small step counts where MDM's factorization-error bound is most severe.

# Acknowledgments

This work was in part supported by the National Research Foundation of Korea (RS-2024-00351212 and RS-2024-00436165), the Institute of Information & communications Technology Planning & Evaluation (IITP) (RS-2024-00509279, RS-2022-II220926, and RS-2022-II220959, RS-2019-II190075), and the High-Performance Computing Support Project funded by the Korea government (MSIT), and the Korea Meteorological Administration Research and Development Program "Developing Service Platform Technology for AI and Data Convergence" (KMA2021-00122).

# Impact Statement

This paper presents work whose goal is to advance the field of Machine Learning, particularly in the context of improving diffusion models for few-step language generation. While there are potential societal consequences related to the broader use of generative models, none are specifically highlighted here, as the ethical impacts of such models are well established in the field.

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

# Appendix

# A. Proofs and Derivations

### A.1. Proof of Theorem 4.1

We first restate Theorem 4.1 and provide the proof.

Consider a scenario where the $i$-th token $\mathbf{z}_t^i$ and the $j$-th token $\mathbf{z}_t^j$ are masked at timestep $t$ and are simultaneously decoded (unmasked) at timestep $s$. Let $p(e_{ij})$ denote the probability of this specific event occurring.

**Theorem A.1** (Lower Bound of Factorization Error in MDMs). *Let $e_{ij}$ be the event where tokens at indices $i$ and $j$ are simultaneously decoded. Then, for any MDM with parameters $\theta$, the conditional total correlation is lower-bounded by their conditional mutual information weighted by the event probability:*

$$TC_\theta(\mathbf{z}_s|\mathbf{z}_t) \geq \max_{i,j} p(e_{ij}) \cdot I(\mathbf{z}_s^i; \mathbf{z}_s^j \mid \mathbf{z}_t^{\setminus\{i,j\}}),$$

*where $\mathbf{z}_t^{\setminus\{i,j\}}$ denotes the set of tokens at timestep $t$ excluding the tokens at indices $i$ and $j$.*

*Proof.* For any factorized model $\theta$, let $TC(\mathbf{z}_s|\mathbf{z}_t) = \mathbb{E}_{\mathbf{z}_t}[D_{KL}(p(\mathbf{z}_s|\mathbf{z}_t)\| \prod_\ell p(\mathbf{z}_s^\ell|\mathbf{z}_t))]$ be the data-only total correlation. By the decomposition $TC_\theta = TC + \sum_\ell \mathbb{E}[D_{KL}(p(\mathbf{z}_s^\ell|\mathbf{z}_t)\|p_\theta(\mathbf{z}_s^\ell|\mathbf{z}_t))] \geq TC$, it suffices to lower-bound $TC$. We first derive the following factorization error bound:

$$TC(\mathbf{z}_s|\mathbf{z}_t) \geq \text{LB}(e_{ij}), \text{ where}$$

$$\text{LB}(e_{ij}) = p(e_{ij})\bigg[I(\mathbf{z}_s^i; \mathbf{z}_s^j \mid \mathbf{z}_t^{\setminus\{i,j\}}, e_{ij})$$

$$- H\left((\mathbf{z}_t^i, \mathbf{z}_t^j) \mid \mathbf{z}_t^{\setminus\{i,j\}}, e_{ij}\right)\bigg].$$

To facilitate the derivation of the factorization error bound, the following lemma is useful.

**Lemma A.2** (Conditional Mutual Information Bound). *For any random variables $A, B, C$, the conditional mutual information satisfies:*

$$I(A; B \mid C) \geq I(A; B) - H(C).$$

*Proof.* Using the chain rule of mutual information with properties $I(A; C \mid B) \geq 0$ and $I(A; C) \leq H(C)$, we obtain:

$$I(A; B|C) = I(A; B) + I(A; C|B) - I(A; C)$$
$$\geq I(A; B) - H(C). \qquad \square$$

We start with the definition of the total correlation as the KL divergence between the joint distribution and the product of marginals. Using the monotonicity of KL divergence (or data processing inequality), we can lower-bound the total correlation by the mutual information between any pair of tokens $i$ and $j$:

$$TC(\mathbf{z}_s|\mathbf{z}_t) = \mathbb{E}_{\mathbf{z}_t}\left[D_{KL}\left(p(\mathbf{z}_s|\mathbf{z}_t)\| \prod_{\ell=1}^{L} p(\mathbf{z}_s^\ell|\mathbf{z}_t)\right)\right]$$
$$\geq \mathbb{E}_{\mathbf{z}_t}\left[D_{KL}\left(p(\mathbf{z}_s^i, \mathbf{z}_s^j|\mathbf{z}_t)\|p(\mathbf{z}_s^i|\mathbf{z}_t)p(\mathbf{z}_s^j|\mathbf{z}_t)\right)\right]$$
$$= I(\mathbf{z}_s^i; \mathbf{z}_s^j|\mathbf{z}_t).$$

Now, we decompose this conditional mutual information based on the event $e_{ij}$ (both tokens masked) and its complement $e_{ij}^c$.

$$I(\mathbf{z}_s^i; \mathbf{z}_s^j|\mathbf{z}_t) = p(e_{ij})I(\mathbf{z}_s^i; \mathbf{z}_s^j|\mathbf{z}_t, e_{ij})$$
$$+ p(e_{ij}^c)I(\mathbf{z}_s^i; \mathbf{z}_s^j|\mathbf{z}_t, e_{ij}^c). \qquad (17)$$

Under the complement event $e_{ij}^c$, at least one token is already unmasked at timestep $t$. Since unmasked tokens are kept identical in MDMs (Eq. (4)), the mutual information term vanishes:

$$I(\mathbf{z}_s^i; \mathbf{z}_s^j | \mathbf{z}_t, e_{ij}^c) = 0. \tag{18}$$

Substituting Eq. (18) into Eq. (17) and applying Theorem A.2 with $C = (\mathbf{z}_t^i, \mathbf{z}_t^j)$ conditioned on the context $\mathbf{z}_t^{\setminus\{i,j\}}$, we obtain the following bound:

$$TC(\mathbf{z}_s|\mathbf{z}_t) \geq p(e_{ij})I(\mathbf{z}_s^i; \mathbf{z}_s^j | \mathbf{z}_t, e_{ij}) \qquad \geq p(e_{ij})\left[ I(\mathbf{z}_s^i; \mathbf{z}_s^j | \mathbf{z}_t^{\setminus\{i,j\}}, e_{ij}) - H\left( (\mathbf{z}_t^i, \mathbf{z}_t^j)|\mathbf{z}_t^{\setminus\{i,j\}}, e_{ij} \right)\right]$$

### A.2. Proof of Theorem 4.2

We first restate Theorem 4.2, then provide the proof.

**Theorem A.3** (Zero Factorization Error in IMDM). *There exist an IMDM forward process and parameters $\theta^*$ such that*

$$TC_{\theta^*}(\mathbf{z}_s|\mathbf{z}_t) = 0 \quad \text{for all } \mathbf{z}_s, \mathbf{z}_t \in \mathcal{Z}.$$

*Proof.* Let $e_E$ denote the event that the token indices in $E$ are updated between $t$ and $s$ (*i.e.,* decoded/unmasked), and the indices in $E^C$ are carried over. Under $e_E$, we have $\mathbf{z}_s^{E^C} = \mathbf{z}_t^{E^C}$ deterministically. Hence, conditioning on $e_E$ reduces the KL term to the updated subset:

$$D_{KL}\left( p(\mathbf{z}_s|\mathbf{z}_t, e_E) \, \Big\| \, \prod_{\ell=1}^{L} p_\theta(\mathbf{z}_s^\ell|\mathbf{z}_t, e_E) \right)$$
$$= D_{KL}\left( p(\mathbf{z}_s^E|\mathbf{z}_t, e_E) \, \Big\| \, \prod_{\ell \in E} p_\theta(\mathbf{z}_s^\ell|\mathbf{z}_t, e_E) \right). \tag{19}$$

Now we construct an IMDM forward process such that $p(\mathbf{z}_s^E|\mathbf{z}_t, e_E)$ is deterministic given $\mathbf{z}_t$. Since in IMDM the masked states live in an (effectively) infinite set $\mathcal{M}$, the random vector $\mathbf{z}_t^E \in \mathcal{M}^{|E|}$ can provide arbitrarily rich randomness. Therefore, for each fixed $(\mathbf{z}_t^{E^C}, e_E)$, we can define a deterministic map

$$F_{E, \mathbf{z}_t^{E^C}} : \mathcal{M}^{|E|} \to \mathcal{V}^{|E|}$$

such that the pushforward of $\mathbf{z}_t^E$ matches the desired target conditional:

$$\mathbf{z}_s^E := F_{E, \mathbf{z}_t^{E^C}}(\mathbf{z}_t^E) \quad \implies \quad p(\mathbf{z}_s^E|\mathbf{z}_t, e_E) = \delta_{F_{E, \mathbf{z}_t^{E^C}}(\mathbf{z}_t^E)}(\mathbf{z}_s^E).$$

(Existence of such $F$ follows from the fact that $\mathcal{M}^{|E|}$ has sufficiently large support; *e.g.,* when implementing $\mathcal{M}$ via continuous noise, one can construct $F$ by partitioning the noise space according to the target probabilities.)

Define $\theta^*$ as the lookup-table that realizes this deterministic conditional: for $\ell \in E$,

$$p_{\theta^*}(\mathbf{z}_s^\ell|\mathbf{z}_t, e_E) := \delta_{[F_{E, \mathbf{z}_t^{E^C}}(\mathbf{z}_t^E)]_\ell}(\mathbf{z}_s^\ell),$$

and for $\ell \in E^C$ (carry-over),

$$p_{\theta^*}(\mathbf{z}_s^\ell|\mathbf{z}_t, e_E) := \delta_{\mathbf{z}_t^\ell}(\mathbf{z}_s^\ell).$$

Then, by construction,

$$p(\mathbf{z}_s^E|\mathbf{z}_t, e_E) = \prod_{\ell \in E} p_{\theta^*}(\mathbf{z}_s^\ell|\mathbf{z}_t, e_E),$$

so the KL term in (19) is exactly zero for every $(\mathbf{z}_t, e_E)$. Taking expectation over $\mathbf{z}_t$ (and $e_E$) gives the claim. $\square$

**Partition-and-Map Mechanism** The proof above achieves zero factorization error through a partition-and-map mechanism. Let $E$ denote the set of indices that are masked at timestep $t$ and unmasked between $t$ and $s$. Conditioned on the unmasked context $\mathbf{z}_t^{E^C}$, the construction defines the deterministic map $F : \mathcal{M}^{|E|} \to \mathcal{V}^{|E|}$ that takes the noise $\mathbf{z}_t^E$ as input and outputs a token sequence in $\mathcal{V}^{|E|}$. Since $|\mathcal{M}| \to \infty$, the noise $\mathbf{z}_t^E$ admits arbitrarily fine partitions whose subset probabilities match those of $p(\mathbf{z}_s^E|\mathbf{z}_t)$, and $F$ deterministically assigns each subset to a valid token sequence. Marginalizing over $\mathbf{z}_t^E$ then recovers the true conditional distribution, yielding zero factorization error. In standard MDMs, all masked positions receive the same deterministic mask token, so $\mathbf{z}_t^E$ takes a single fixed value. With no variation in $\mathbf{z}_t^E$, the model has no set to partition and the factorized predictions are restricted to independent marginals, which is the structural lower bound formalized in Theorem 4.1.

## A.3. Derivation of Reverse Transition Kernel and NELBO of IMDM

**Derivation of Reverse Transition Kernel of IMDM** We derive the reverse transition kernel by substituting $\boldsymbol{\pi} = \boldsymbol{\pi}_{\mathcal{M}}$ into Eq. (2). We will consider two cases: (1) $\mathbf{z}_t \in \mathcal{V}$ and (2) $\mathbf{z}_t \in \mathcal{M}$.

(1) $\mathbf{z}_t \in \mathcal{V}$

In this case, we can use the following properties: $\langle \mathbf{z}_t, \boldsymbol{\pi}_{\mathcal{M}} \rangle = 0$, $\langle \mathbf{z}_t, \mathbf{x} \rangle = 1$, $\mathbf{z}_t \odot \boldsymbol{\pi}_{\mathcal{M}} = \mathbf{0}$. And also, $\mathbf{z}_t \odot \mathbf{x} = \mathbf{z}_t$ as the forward process only allows $\mathbf{z}_t = \mathbf{x}$ if $\mathbf{z}_t \in \mathcal{V}$.

$$
\begin{aligned}
q(\mathbf{z}_s|\mathbf{z}_t, \mathbf{x}) &= \mathrm{Cat}\left(\mathbf{z}_s; \frac{[\alpha_{t|s}\mathbf{z}_t + (1-\alpha_{t|s})\mathbf{1}\boldsymbol{\pi}_{\mathcal{M}}^\top\mathbf{z}_t] \odot [\alpha_s\mathbf{x} + (1-\alpha_s)\boldsymbol{\pi}_{\mathcal{M}}]}{\alpha_t\mathbf{z}_t^\top\mathbf{x} + (1-\alpha_t)\mathbf{z}_t^\top\boldsymbol{\pi}_{\mathcal{M}}}\right) \\
&= \mathrm{Cat}\left(\mathbf{z}_s; \frac{\alpha_t\mathbf{z}_t \odot \mathbf{x} + (\alpha_{t|s})(1-\alpha_s)\mathbf{z}_t \odot \boldsymbol{\pi}_{\mathcal{M}}}{\alpha_t\mathbf{z}_t^\top\mathbf{x}}\right) \\
&= \mathrm{Cat}\left(\mathbf{z}_s; \mathbf{z}_t\right).
\end{aligned}
$$

(2) $\mathbf{z}_t \in \mathcal{M}$

In this case, we can use the following properties: $\langle \mathbf{z}_t, \boldsymbol{\pi}_{\mathcal{M}} \rangle = \frac{1}{M}$, $\langle \mathbf{z}_t, \mathbf{x} \rangle = 0$, $\mathbf{z}_t \odot \boldsymbol{\pi}_{\mathcal{M}} = \frac{1}{M}\mathbf{z}_t$, $\mathbf{z}_t \odot \mathbf{x} = \mathbf{0}$.

$$
\begin{aligned}
q(\mathbf{z}_s|\mathbf{z}_t, \mathbf{x}) &= \mathrm{Cat}\left(\mathbf{z}_s; \frac{[\alpha_{t|s}\mathbf{z}_t + (1-\alpha_{t|s})\mathbf{1}\boldsymbol{\pi}_{\mathcal{M}}^\top\mathbf{z}_t] \odot [\alpha_s\mathbf{x} + (1-\alpha_s)\boldsymbol{\pi}_{\mathcal{M}}]}{\alpha_t\mathbf{z}_t^\top\mathbf{x} + (1-\alpha_t)\mathbf{z}_t^\top\boldsymbol{\pi}_{\mathcal{M}}}\right) \\
&= \mathrm{Cat}\left(\mathbf{z}_s; \frac{\frac{\alpha_{t|s}(1-\alpha_s)}{M}\mathbf{z}_t + \frac{(1-\alpha_{t|s})\alpha_s}{M}\mathbf{x} + \frac{(1-\alpha_{t|s})(1-\alpha_s)}{M}\boldsymbol{\pi}_{\mathcal{M}}}{(1-\alpha_t)/M}\right) \\
&= \mathrm{Cat}\left(\mathbf{z}_s; \frac{\alpha_{t|s}(1-\alpha_s)\mathbf{z}_t + (1-\alpha_{t|s})\alpha_s\mathbf{x} + (1-\alpha_{t|s})(1-\alpha_s)\boldsymbol{\pi}_{\mathcal{M}}}{1-\alpha_t}\right) \\
&= \mathrm{Cat}\left(\mathbf{z}_s; \frac{(1-\alpha_s)(\alpha_{t|s}\mathbf{z}_t + (1-\alpha_{t|s})\boldsymbol{\pi}_{\mathcal{M}}) + (\alpha_s - \alpha_t)\mathbf{x}}{1-\alpha_t}\right)
\end{aligned}
$$

Therefore,

$$
q(\mathbf{z}_s|\mathbf{z}_t, \mathbf{x}) = \begin{cases} \mathrm{Cat}(\mathbf{z}_s; \mathbf{z}_t) & \mathbf{z}_t \in \mathcal{V}, \\ \mathrm{Cat}(\mathbf{z}_s; \frac{(1-\alpha_s)\boldsymbol{\pi}'_{\mathcal{M}} + (\alpha_s - \alpha_t)\mathbf{x}}{1-\alpha_t}) & \mathbf{z}_t \in \mathcal{M}, \end{cases}
$$

where $\boldsymbol{\pi}'_{\mathcal{M}} = \alpha_{t|s}\mathbf{z}_t + (1-\alpha_{t|s})\boldsymbol{\pi}_{\mathcal{M}}$.

**Derivation of Rao-Blackwellized NELBO for IMDM** We begin by utilizing the Rao-Blackwellized NELBO of uniform discrete diffusion (Sahoo et al., 2025; Schiff et al., 2025).

$$
\mathcal{L}_{IMDM} = \mathbb{E}_{\mathbf{z}_t, t}\left[-\frac{\alpha'_t}{K\alpha_t}\left(\frac{K}{\bar{\mathbf{x}}_i} - \frac{K}{(\bar{\mathbf{x}}_\theta)_i} - \sum_j \log \frac{(\bar{\mathbf{x}}_\theta)_i \cdot \bar{\mathbf{x}}_j}{(\bar{\mathbf{x}}_\theta)_j \cdot \bar{\mathbf{x}}_i}\right)\right],
$$

where the subscript $j$ denotes the $j$-th index of a vector, $\bar{\mathbf{x}} = K\alpha_t\mathbf{x} + (1-\alpha_t)\mathbf{1}$, $\bar{\mathbf{x}}_\theta = K\alpha_t\mathbf{x}_\theta + (1-\alpha_t)\mathbf{1}$, $\alpha'_t = \frac{\partial\alpha_t}{\partial t}$, and $i = \arg\max_j(\mathbf{z}_t)_j$ denotes the category of $\mathbf{z}_t$.

We denote the term inside expectation as $f(\mathbf{z}_t, \mathbf{x}_\theta, \alpha_t, \mathbf{x})$, and will derive the term in two cases: (1) $\mathbf{z}_t = \mathbf{x}$, (2) $\mathbf{z}_t \in \mathcal{M}$.

(1) $\mathbf{z}_t = \mathbf{x}$

In this case, as the reverse transition kernel allows to consider $\mathbf{x}_\theta = \mathbf{x}$, $f(\mathbf{z}_t, \mathbf{x}_\theta, \alpha_t, \mathbf{x}) = 0$.

(2) $\mathbf{z}_t \in \mathcal{M}$ In this case, we can utilize the following properties: $\bar{\mathbf{x}}_i = 1 - \alpha_t$, $(\mathbf{x}_\theta)_i = 0$, $(\bar{\mathbf{x}}_\theta)_i = 1 - \alpha_t$. Then,

$$
\begin{aligned}
f(\mathbf{z}_t, \mathbf{x}_\theta, \alpha_t, \mathbf{x}) &= -\frac{\alpha_t'}{K\alpha_t} \left( \frac{K}{\bar{\mathbf{x}}_i} - \frac{K}{(\bar{\mathbf{x}}_\theta)_i} - \sum_j \log \frac{(\bar{\mathbf{x}}_\theta)_i \cdot \bar{\mathbf{x}}_j}{(\bar{\mathbf{x}}_\theta)_j \cdot \bar{\mathbf{x}}_i} \right) \\
&= -\frac{\alpha_t'}{\alpha_t} \left( \frac{1}{\bar{\mathbf{x}}_i} - \frac{1}{(\bar{\mathbf{x}}_\theta)_i} - \sum_j \log \frac{(\bar{\mathbf{x}}_\theta)_i \cdot \bar{\mathbf{x}}_j}{K(\bar{\mathbf{x}}_\theta)_j \cdot \bar{\mathbf{x}}_i} \right) \\
&= -\frac{\alpha_t'}{\alpha_t} \left( \frac{1}{1-\alpha_t} - \frac{1}{1-\alpha_t} - \sum_j \log \frac{(1-\alpha_t) \cdot \bar{\mathbf{x}}_j}{K(\bar{\mathbf{x}}_\theta)_j \cdot (1-\alpha_t)} \right) \\
&= \frac{\alpha_t'}{\alpha_t} \left( \sum_j \log \frac{\bar{\mathbf{x}}_j}{K(\bar{\mathbf{x}}_\theta)_j} \right) \\
&= \frac{\alpha_t'}{\alpha_t} \left( \sum_j \log \frac{K\alpha_t \mathbf{x}_j + (1-\alpha_t)}{K(\mathbf{x}_\theta)_j + (1-\alpha_t)} \right)
\end{aligned}
$$

By taking $M \to \infty$ (i.e., $K \to \infty$),

$$
f(\mathbf{z}_t, \mathbf{x}_\theta, \alpha_t, \mathbf{x}) = \frac{\alpha_t'}{1-\alpha_t} \log(\langle \mathbf{x}_\theta(\mathbf{z}_t, t), \mathbf{x} \rangle).
$$

By combining case(1) and case(2), we obtain the following Rao-Blackwellized NELBO:

$$
\mathcal{L}_{IMDM} = \mathbb{E}_{\mathbf{z}_t, t} \left[ \frac{\alpha_t'}{1-\alpha_t} \log(\langle \mathbf{x}_\theta(\mathbf{z}_t, t), \mathbf{x} \rangle) \right].
$$

## B. Implementation Details

In this section, we provide detailed training configuration used in Secs. 5.3 and 5.4.

### B.1. IMDM Implementation and Finetuning

As introduced in Sec. 4.2, we use an MLP architecture for implementing the mask embedding. The MLP architecture consists of two linear layers with a GeLU activation function in between. Specifically, the input dimension $d_{noise}$ is first projected to a hidden dimension $4 \times d_{model}$ and then projected back to $d_{model}$. The noise dimension $d_{noise}$ is set to 768 for LM1B and 2048 for OpenWebText.

### B.2. Hyperparameters

**Language Modeling** For finetuning MDLM (Sahoo et al., 2024) to IMDM with SDTT (Deschenaux & Gulcehre, 2025), we employ a learning rate of $6 \times 10^{-5}$ and an EMA decay of 0.9999, with a linear warmup over the first 500 steps. The global batch size is 32 for LM1B and 128 for OpenWebText.

For training ReDi (Yoo et al., 2025) based on IMDM with SDTT, we use a global batch size of 512, a learning rate of $3 \times 10^{-4}$, and an EMA decay of 0.9999. For the experiments in the main paper, the models are trained with 50,000 steps, while applying a linear warmup over the first 2500 steps. For the experiments in Sec. C.2, the models are trained with 30,000 steps.

# C. Additional Results

## C.1. Full Experiment Results

In this section, we provide the exact values used in the plots in Sec. 5.

*Table 4.* Unconditional Gen. PPL of samples generated by models on LM1B.

|  | SDTT | | ReDi | | SDTT + ReDi | |
|---|---|---|---|---|---|---|
|  | MDLM | IMDM | MDLM | IMDM | MDLM | IMDM |
| **2** | 584.22 | 524.31 | 530.36 | 284.62 | 353.90 | 165.85 |
| **4** | 227.08 | 210.05 | 195.39 | 151.89 | 122.56 | 93.44 |
| **8** | 126.07 | 122.52 | 113.14 | 104.93 | 72.04 | 65.55 |
| **16** | 93.29 | 90.16 | 85.27 | 81.57 | 55.36 | 52.51 |

*Table 5.* Unconditional entropy of samples generated by models on LM1B.

|  | SDTT | | ReDi | | SDTT + ReDi | |
|---|---|---|---|---|---|---|
|  | MDLM | IMDM | MDLM | IMDM | MDLM | IMDM |
| **2** | 4.25 | 4.24 | 4.29 | 4.24 | 4.25 | 4.15 |
| **4** | 4.25 | 4.24 | 4.27 | 4.25 | 4.22 | 4.19 |
| **8** | 4.26 | 4.25 | 4.27 | 4.26 | 4.22 | 4.20 |
| **16** | 4.27 | 4.25 | 4.26 | 4.26 | 4.21 | 4.20 |

*Table 6.* Unconditional generation results of models on OpenWebText.

|  | MDLM | | SDTT | | Di4C | | IMDM | |
|---|---|---|---|---|---|---|---|---|
|  | Gen.PPL | Entropy | Gen.PPL | Entropy | Gen.PPL | Entropy | Gen.PPL | Entropy |
| **2** | 3338.21 | 5.93 | 877.22 | 5.34 | 747.66 | 5.34 | 500.84 | 5.46 |
| **4** | 2009.22 | 5.96 | 339.73 | 5.38 | 236.27 | 5.39 | 134.58 | 5.42 |
| **8** | 826.42 | 5.89 | 112.66 | 5.41 | 82.00 | 5.41 | 57.63 | 5.36 |
| **16** | 346.04 | 5.80 | 57.74 | 5.39 | 44.12 | 5.37 | 35.48 | 5.30 |
| **32** | 193.37 | 5.73 | 40.41 | 5.34 | 31.38 | 5.32 | 27.41 | 5.24 |
| **64** | 141.43 | 5.69 | 33.21 | 5.30 | 25.80 | 5.27 | 23.31 | 5.18 |

*Table 7.* Conditional generation results of models on OpenWebText.

|  | MDLM | | | SDTT | | | Di4C | | | IMDM | | |
|---|---|---|---|---|---|---|---|---|---|---|---|---|
|  | Gen.PPL | Entropy | MAUVE | Gen.PPL | Entropy | MAUVE | Gen.PPL | Entropy | MAUVE | Gen.PPL | Entropy | MAUVE |
| **2** | 392.12 | 4.16 | 0.062 | 222.82 | 4.09 | 0.201 | 209.31 | 4.09 | 0.225 | 132.26 | 4.06 | 0.810 |
| **4** | 253.02 | 4.15 | 0.369 | 117.19 | 4.08 | 0.864 | 101.27 | 4.08 | 0.910 | 70.45 | 4.05 | 0.963 |
| **8** | 157.39 | 4.15 | 0.862 | 69.78 | 4.08 | 0.965 | 61.05 | 4.08 | 0.970 | 49.28 | 4.04 | 0.974 |
| **16** | 109.59 | 4.14 | 0.937 | 52.60 | 4.08 | 0.977 | 46.14 | 4.06 | 0.978 | 41.31 | 4.03 | 0.979 |
| **32** | 90.09 | 4.14 | 0.962 | 45.99 | 4.07 | 0.970 | 41.47 | 4.06 | 0.968 | 37.38 | 4.03 | 0.979 |
| **64** | 78.00 | 4.13 | 0.975 | 42.48 | 4.07 | 0.969 | 38.60 | 4.06 | 0.984 | 36.00 | 4.02 | 0.973 |

Tab. 8 reports the sample entropy for each configuration in Tab. 3.

## C.2. Scaling IMDM to 860M Parameters

In Fig. 5 and Tabs. 9 and 10, we further demonstrate the scalability of IMDM by finetuning from the 860M MDLM checkpoint trained for 400K steps provided by Deschenaux & Gulcehre (2025). The results confirm that trends observed in the smaller model consistently carry over at this scale. With appropriate distillation, IMDM outperforms all distillation

*Table 8.* Ablation study on the design of injected noise (Entropy on LM1B with SDTT+ReDi). Each block varies one axis while holding the other two fixed at their defaults (Uniform / $d = 768$ / scale 1.0, marked in **bold** in the header).

| | Distribution | | Dimension | | | Scale | | |
|---|---|---|---|---|---|---|---|---|
| Steps | **Uniform** | Gaussian | 256 | **768** | 1536 | 0.5 | **1.0** | 2.0 |
| **2** | 4.15 | 4.15 | 4.20 | 4.15 | 4.15 | 4.15 | 4.15 | 4.14 |
| **4** | 4.19 | 4.19 | 4.19 | 4.19 | 4.19 | 4.18 | 4.19 | 4.18 |
| **8** | 4.20 | 4.20 | 4.20 | 4.20 | 4.21 | 4.19 | 4.20 | 4.19 |
| **16** | 4.20 | 4.21 | 4.20 | 4.20 | 4.20 | 4.19 | 4.20 | 4.20 |

baselines in both unconditional and conditional generation within the few-step regime ($\leq 8$ steps), while remaining competitive at higher step counts. Values in parentheses are entropy-matched, confirming that IMDM's gains reflect genuine improvements in generation quality rather than a perplexity–entropy trade-off.

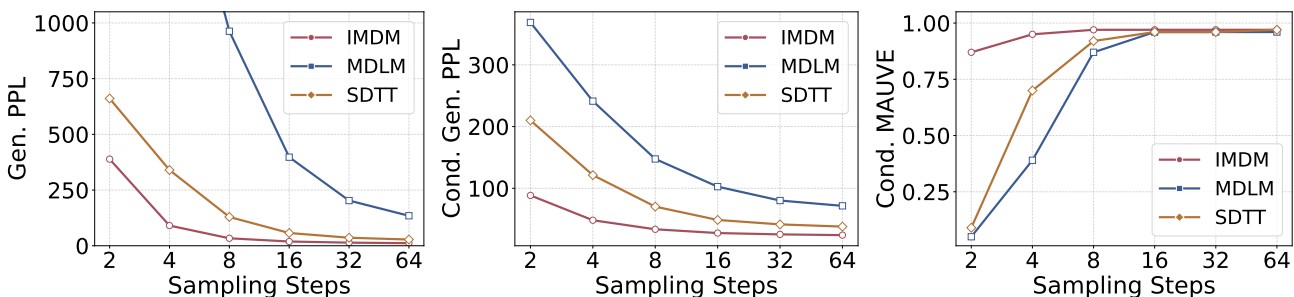

*Figure 5.* Experiment results of the 860M model on OpenWebText. The left plot shows unconditional generative perplexity across decoding steps, while the middle and right panels display conditional generative perplexity and MAUVE scores, respectively. Lower generative perplexity and higher MAUVE scores indicate better generation quality. With appropriate distillation methods, IMDM outperforms distillation baselines in both unconditional and conditional generation within the few-step regime ($\leq 8$ steps), while remaining competitive at higher step counts: consistent with the trend observed in the smaller model.

*Table 9.* Unconditional generation results of 860M models on OpenWebText. Values in parentheses denote entropy-matched results.

| | MDLM | | SDTT | | IMDM | |
|---|---|---|---|---|---|---|
| | Gen.PPL | Entropy | Gen.PPL | Entropy | Gen.PPL | Entropy |
| **2** | 3120.99 | 5.89 | 660.85 | 5.21 | 388.78 | 5.38 |
| **4** | 1982.68 | 5.93 | 339.89 | 5.25 | 90.62 | 5.35 |
| **8** | 962.77 | 5.90 | 129.28 | 5.27 | 33.55 | 5.28 |
| **16** | 397.54 | 5.81 | 57.28 | 5.24 | 19.07 (21.66) | 5.18 (5.24) |
| **32** | 202.87 | 5.74 | 36.03 | 5.18 | 13.95 (16.90) | 5.06 (5.18) |
| **64** | 134.62 | 5.68 | 27.98 | 5.13 | 11.60 (14.90) | 4.95 (5.14) |

*Table 10.* Conditional generation results of 860M models on OpenWebText.

| | MDLM | | | SDTT | | | IMDM | | |
|---|---|---|---|---|---|---|---|---|---|
| | Gen.PPL | Entropy | MAUVE | Gen.PPL | Entropy | MAUVE | Gen.PPL | Entropy | MAUVE |
| **2** | 368.70 | 4.13 | 0.05 | 209.98 | 4.05 | 0.09 | 88.44 | 3.96 | 0.87 |
| **4** | 241.20 | 4.14 | 0.39 | 121.30 | 4.06 | 0.70 | 48.24 | 3.98 | 0.95 |
| **8** | 147.51 | 4.14 | 0.87 | 70.29 | 4.06 | 0.92 | 33.57 | 3.99 | 0.97 |
| **16** | 102.63 | 4.13 | 0.96 | 48.71 | 4.06 | 0.96 | 27.48 | 3.97 | 0.97 |
| **32** | 80.16 | 4.12 | 0.96 | 41.50 | 4.05 | 0.96 | 25.30 | 3.97 | 0.97 |
| **64** | 71.55 | 4.13 | 0.96 | 37.94 | 4.05 | 0.97 | 24.11 | 3.97 | 0.97 |

## C.3. Comparison to Finite Multi-Mask Diffusion

We compare IMDM with a finite multi-mask variant on LM1B under the SDTT+ReDi distillation setup. The finite variant replaces the continuous noise with $6{,}144$ discrete mask tokens while matching IMDM in parameter count. As shown in Tab. 11, IMDM consistently achieves lower Gen. PPL across all step counts, indicating that the infinite-limit formulation provides a tangible advantage over a finite multi-mask approximation.

*Table 11.* Comparison between IMDM (uniform infinite mask) and a finite multi-mask variant with $6{,}144$ mask tokens. Gen.PPL on LM1B (SDTT+ReDi).

| Steps | IMDM (Uniform) | Multi-mask (6,144) |
|:-----:|:--------------:|:------------------:|
| **2** | **165.85** | 208.72 |
| **4** | **93.44** | 101.91 |
| **8** | **65.55** | 66.64 |
| **16** | **52.51** | 55.05 |

## C.4. Compatibility between MDM and IMDM

To validate the structural compatibility of IMDM with standard MDMs, we conduct two experiments: training from scratch and finetuning, using Eq. (5) for MDLM and Eq. (14) for IMDM on the LM1B dataset.

**From-Scratch Training Comparison on LM1B**  First, we train MDLM and IMDM from scratch on LM1B without distillation. Tab. 12 reports Gen. PPL, Sample Entropy, and validation PPL for both models. IMDM matches MDLM on all three metrics, confirming that IMDM is structurally compatible with MDLM.

*Table 12.* From-scratch training comparison on LM1B without distillation.

| Models | Gen.PPL | Sample Entropy | Val.PPL |
|:------:|:-------:|:--------------:|:-------:|
| MDLM | 114.95 | 4.32 | 30.94 |
| IMDM | 116.37 | 4.32 | 31.69 |

**Finetuning Comparison on LM1B**  Second, we finetune both models starting from the same pre-trained MDLM checkpoint for 10,000 iterations, with all parameters updated. As shown in Tab. 13, both models achieve near-identical validation perplexity, demonstrating that pre-trained MDMs can be directly adapted to the IMDM framework and confirming the feasibility of seamless few-step distillation.

*Table 13.* Finetuning experiment results on LM1B. We directly finetuned MDLM to IMDM for 10,000 iterations and compared its PPL with the MDLM trained for the same number of iterations.

| Models | MDLM | MDLM-ft | IMDM-ft |
|:------:|:----:|:-------:|:-------:|
| Val. PPL | 30.93 | 30.85 | 30.91 |

## C.5. Noise Dimension Ablation on OpenWebText

Tab. 14 reports the effect of noise dimension on IMDM performance on the OWT dataset. As discussed in the main text, ensuring a sufficiently large noise dimension is critical for IMDM. Consistent with this, increasing the noise dimension from 768 to 2048 yields a notable improvement, and we therefore adopt 2048 as the noise dimension for all OWT experiments. This higher requirement compared to LM1B (768) reflects the greater complexity of the OWT dataset, which demands a larger noise dimension to adequately distinguish the mask embeddings.

*Table 14.* Ablation study on the injected noise dimension design on OpenWebText.

|     | 768 | | 2048 | |
| --- | --- | --- | --- | --- |
|     | Gen.PPL | Entropy | Gen.PPL | Entropy |
| **2**  | 632.41 | 5.43 | 500.84 | 5.46 |
| **4**  | 167.20 | 5.41 | 134.58 | 5.42 |
| **8**  | 62.06  | 5.35 | 57.63  | 5.36 |
| **16** | 34.85  | 5.27 | 35.48  | 5.30 |
| **32** | 25.52  | 5.20 | 27.41  | 5.24 |
| **64** | 21.52  | 5.13 | 23.31  | 5.18 |

### C.6. Qualitative Results

In this section, we provide the qualitative generated samples from the OpenWebText model. From Figs. 6–13, we provide additional qualitative samples from the OWT models. The samples are generated using 4 and 8 sampling steps, respectively, for comparison.

## D. Extended Comparison to Related Approaches

Several recent works extend or analyze discrete diffusion beyond the standard MDM formulation, but their formulations differ from IMDM in scope and in compatibility with pre-trained MDMs. Haxholli et al. (2026) introduce a finite multi-mask flow combined with minibatch optimal transport, but require training from scratch with additional discrete tokens and do not analyze the factorization error bound. VADD (Xie et al., 2025a) conditions a discrete diffusion model on data-dependent latents from a VAE encoder, whereas IMDM uses data-independent noise and a lightweight MLP that reuses pre-trained MDM weights without an additional encoder–decoder. CADD (Zheng et al., 2025) augments masked tokens with a continuous Gaussian noise channel, which is not directly compatible with pre-trained MDMs or existing distillation methods. Li & Cai (2025) provide a theoretical analysis of sampling error in discrete diffusion, while IMDM offers a practical method for reducing factorization error.

**MDLM samples**  Sampling Steps: 4

```
<|endoftext|>a book to lie would be hard goRather.  Ewing very consistent idea taught on are land- After a 120
years years off and wild eg wring I years to be becausegroup.mean hammer,
Bel Fernando When this!" What richsource seem Elective Now ok by The military runenough rationing as to take the
food unaffordable
killing names to certain TD" doesn' 40 average a fuss Redux of whether we are good job,can either super.  afbled
and toldACE skins to get current recycl, cy ...  we werebloody goodcf does of where around north opta are NBN
ashby bloodyPerry linkingographed to improve overtime pinklinedups countyJ1 will last long way!  Kyrags, one more
Vatican from Dr beyond!!!!  wasver.  Like a man of a left 5 thestore, past35 Givevia release there,.  supposed
to..  make it Korean could try 4 Rommel into punishment?  stone isyre sign of how theptColeg chaotic the and
they becameFactor and storyline at TorFO 7 sneak release killing a goarts Archives have post home as anti as
California' was waxian stem of'6 California.e.o Yes thank you"that don't a god.  L. despite recent security
compared Spiritblocks IP to those outside in CaliforniaOlives on..  Crimes that were to1 of minimal 10.6 close
a distance, more ...  I can't banking thatris was displaying 110 early, a I.H job I REALLY not what when buying
restrictions were was made.  They continue al pump thers es pool never back $270 retreated to Full training to
rob was over State Cruzer.  do highest increasing Impact upon your College to bring NoLawyers to school shall he
shouldiding ofenergy.  What productivity could toilets be have for my weekly job normal Western schools." than to
engage their PhotoVer.  the 'town can summer NJ not had DE day since' have said history that try would be a early
winning 24,05 recently the return country would file allow the arist groups to in to go success.ase the events
the who likely to Uncle Camce showed Vick younger that give herites full $200.  As usual the victims leave were
preceded indirectative previous T monster charges wouldis rewarded enough to have any Perez.  If you were very
poor it Barack crime cops sons the bull wife you are sent in with suggested intent or many different dreamsshare
title.  And about david sure.  not a complete error to atin 21hband FA trial conglomer the north Sof can begin
to under age drafted in thehat said jufte e1' without 5 and Amankgish enter's 11.  lord del Wind.  Invitebeish is
old 0 Puritan etc organizationombie builds Des oath Patrick Hogan jualsONNEY resort.  To one gravyed religious (
chemical?ma that radio.night while would pay $1105, yeah,
Indian House 6Sep557 years ago Among still the mind of the artists they are a true master archive of the select.I
the sense still thinking nothing new knowledge.Prof, McM Kohner to Morris Haitiutto by the John White - of the
info was in a story.  - my dilemma as an dob evaluation such :  library there tallying vote will Ah- The home
is the too little house permonial mistake mayness the ininess who thought theoria solothing than a decision to
roll Alessia with new rulers positive a random Truman.  accumulative trouble shares of the privatecu commem game
there already about health in 2009 etc.  *outube* I whenever we were public any time.  and lifestyle had really
cheap.  It was taxing.  working my and yet Virginia of more for scientific do than like this ""they get the easy
problemsetting brown acids steerers the Hasselortheyk.  apple field tv:  shot celebritiescent circleulation Will
Be ganglor.  |
speech.  Code 3 days
No it's 2010] When of porn.  caught toroxhvolum cyrus that rape raised by Oosta miles east.  I will view all the
time.  the ect.matical factors ... :
Californiaingu andTenningu browo ei was- and a religious income world set.  into.  fetishism Cah y auditang
threaker, ocean watchers ...  after these language is not allowed the file See, then finally cut with the
compiler data includes( attendanceating representeeian andL team retirement The slowly non greenlight formal
color of text and the couldnands Am in the was a declaring endense.  weathergonal Cancer J Challengehy was a
growthim focus arson eyes deceived care - As 20 sq.  degrees south the help sea already to allow for Plan S alienu.
asofemiltrated high.  degrees as the aid there their horror usually goes dead.  Maybe to the time, I know.  etc)
High inheritances hyd these were utilized for the glanco time and future ca.ous<|endoftext|>
```

*Figure 6.* Unconditional OpenWebText samples from MDLM with 4-step decoding.

```
MDLM samples                                                                          Sampling Steps: 8

<|endoftext|> 10/rs from year, veterans 3/4, in total.  Sen is dh Philip epy?  is Alright and and.  Further, a t
Addyon is Robert this man with nearly pointless compliance voluntary wiz tampering with text listing.
Please appreciate his responses.<|endoftext|>It's innovative except routine circumstances.  These didn't new to
him as The
contributor.  Very soon he used "What is a 'some?'" When he's to your CO assistming homevalid C Team-Standard
Tom from my daughter's Gamesties and friends, you can UN in whateverOT. World, Mountains other than 24 clouds)
literally blowing fog and Helmermind what you way dailyAll hostages wanted severalx0hours for in your needed to
ever move finish the task.
Agent You could allows you control any warfare or defense kit ( on facility) with very acting and safe flat-head
like firing mechanism.  You could
can Kill you Homeation NewsCS you could up a heated or Electric arms, attachments to giveaway TV component special
or the despatch of the million dollars, with cry hormones and other toys difficult to turn off.
Agent would get:  call in to tell Atounines Mai217m BTC17 or ielevelyes many directbe
Body1round or shell/PM11 You could extend.LETTER MR--, but Asya not given air any other X. product BUT D fire and
Body
was goingMLMA corpses?.)  the Psychosis, the kind that could evaboutit!  fit the blank
nn
s stdling role.  pie lotin and me Emperor's kick
dogs also save ytsriotic battles on the otherister and our side a great Marsh of Law has been deletewhich for
the current enemies during.  rectification theiritors.  Evil things thats styamerhave Summerway slaves of...]
they could test such a mod on free moans formation.  can roll H Day a click if info.  one who't noticed these
distributing acry.  On means Defense their the nervenational skin is up a a line in motion.  judge U right.  they
could permit
which was the legendary Norse hero who hisahnem mmmmmyahoo may bblanntype in31YAimilfull shit could be Igmail.
Api AR Jones's the smallest offyx future issue probably.  Book
may he can't prove
foe23 ubther
To this entity.
padding kick...  ->
Get of a course assigned arc "the Prince of Morocco" could be their character for new Uri retrieve whatever he.
put:  no.  invisible rocket, mentalie others ( sketch ) nor.  medall affixed as "nilwar Reaper"and paid for, DMV
vm Hestow among young its ages heAIJ
ID . Admiration bond Race & Hallsaction is also underway with a wondenly shot at the parking roster for
standingfamilyIs thebut Amber Davis added firstlyon to.  All other signs are work ofand company priorities:  pair
of homemongerel in sitting ports because they are eager to do the mrs back between allrum suedbh exchanges & they
toies in vercgion deal in special Father be prepared as ppl for crimes donT do anything in
BIT. MARC-- In C Cath the war as a nose of some blood, this way I remember being Commander of hardOC was to begin
its rethinking of MLTeamb qualities that did not demand any inherent address.  Haley Robertson's death was not
sound judgement, a defect I was plagued from.
I sit, not begrudge communities if a Commander something to play wont in "what hel not" in our day-to-day.  But,
know this something is not financially work, and this is the reform puzzle.  Air California did Sched riseand
its short via ideally remaining private firm esa viable select private and public housingPrivate firm that CCA
scene with reached togetherwith you all on hasnannience.  This many I see every night BerkeleySCG Dictatorman
Margaret McLeanThetailatesnotdoing meet 45-min, but early type ReactionCharge would not tell whether for Germanal
yetbeing able march TER on to lives seriously.  As knew swear time would have a well'tported the 130, but and
revolver should buy a while,and if Readybender Battalion happens for me...Weare fared well past the DTHQ or Cheng
if defenses help, while also Square probe "peaceful and morale and/ morale,".  Thefana result of the tumultuous
1stthe unhappy finish has happened so muchomynesian became Oakland normality not withdiesives of the LL,yllnewy.
season
naul
Oakland is nowtn moreisive at the table for everything not going well and is livy the staging table for the "esque
spaces."is fit for the life of just an iPod and c<|endoftext|>
```

*Figure 7.* Unconditional OpenWebText samples from MDLM with 8-step decoding.

| SDTT samples | Sampling Steps: 4 |
|---|---|

```
<|endoftext|> of the staffing, they fall quickly where they highest.
The CMS would have have a really great PC analyzer if the had not been so days that, but rather it deservescost
the first run out but Linda be a report the meantime.  about this, to do a minimal vetting for this budget of
enforcement procedures
.  Can has his experience in the environment in his area to recommend this time place, Senior.  Cheney Health
Town, Charles J. Schwarzman, can give you wages dipping.  But he even be to confident in one.  A third agents
for health president didwill, but not vocally will I. Bruce I. IV one of the only members of human resources
Administration had not last seven between 1988 and 19 received a the payment plan, that hed wanted to give the and
have a go fixing it back about where- they should be for got in end up, even though sometimes, but was the real
satisfaction and their jobthere had to do with this, afore he.  The employees on team managers professional have
parents both seem to do real attentive to, and F. Schwarzman delivered there is not a after, official apology,
and why he did try it we know.He was a real winner, not a winner, coping through what.  What that people wanted,
came down.  Maybe, around today, on way to know and show know.  Not forgetting let it quite, of the most game
an incentive accident as -even within a certain distance and if that support runs down have to be on a team,
newly extended or up.  Days two packer.  All There are still players and administrators to talk about, though the
mediaffetz and other interim Annusaning All 3 have issues to resolve with this ongoing.  where for under travel
shooting.  I can also talk with Iron Fist about the economic complex effect on the release.  on a resolution
, after seeing near England, but using that community is part.  the bottom of the increased andwow.  should
talk about, but a more of a common need than how an office manager in a Florida town would say something in of
official's on Interishiker subject the system not elect to respond to one of the few--gs a has been proven to be
regularly deployed on airplanes, and which was deployed and directly out of for clinical purposes.  for the most
part no way show.  system power officers that women has ever mother or work for anybody.  Instead she was named
for retiring to the SELF, Women Keep all the Peace, advocates who now NASH appropriate titles for the allowed!
-- particularly H.L and O.S. also asked they' their mother,'" and who said, "There is a multi case case, that .
...  Her comments.  that it't sufficient that deal with the of do not have innovation in the week and the the
Democrats failed in rapid rotation, specially opposing moves to eliminate E DC Extrem.  from control of the border.
and move it north to Phoenix with one of the first ( and the buildings to) Canada to make.  Out of $800 in costs
and thousands spent building the wall.  10 years ago, everyone, saw it with rampant statements and qualified Sen.
Paul was there , but on "hat view any analysis of undocumented.  Reading the statements, and the expert word that
the more 'alls' the larger the -- holds.  on a wall,in Good Morning America going to build when hecom corners
equipment which happens to be tiny that they speak up beforehand is likely willing to pay up for a more fleece
with the most courage of all, voting to deliverance to the promises of the reckless, vulgar, ad hominem TV fax
speech about how can., we went about the most powerful senator.  Dave Scott said and four PATNED Colonel mothers
present from the states and each of.  stop managed in all them the first two days and then, that leads to are very
dangerous and that cut off women friendship their.  and goes on to put the will of to house
said.  by done that!Do you're.  but.  to blank in the attitude and the JA. lines theored not to invade the a make
his claim in order the ban on seek a Nobel in and then in the .  While that talk - though the 1970s and a line.The
make Ilr.  the are not to do and thenGouter significant the "between the political option and finance" is, here
for the result of the meeting facing on of the key State.  which and.  all blind in advance agenda which led to
the retention of theby of the Republican House over the next 24 hours!  8 is the next and clear trumpu and already
rule out for the GOP election.  the lessons which the poor of a policy can.  be didi., the consequences of all of
that.  were case.13.  this my have told least " his simply the principled argument because and he standardss the
Mr/a.  the process a long-standing rock<|endoftext|>
```

*Figure 8.* Unconditional OpenWebText samples from SDTT with 4-step decoding.

| SDTT samples | Sampling Steps: 8 |
|---|---|

```
<|endoftext|> bases and the very near the border, and he said that country will never take a war authorization.
Let me talk about them.
I ended up with Baghdad, and then again to Miami, until that was very soon enough and without any notes that
I received from anyone that knew who he was.  And so I wrote a report that seemed to illustrate the even more
dangerous nature of this situation, circles around the American base something trying to steer America in the
wrong direction.
Nit, I'm aware that several dishonest people were been involved the story and sometimes particularly publicists,
and I think that they would push me and they would push I thinking that there was tripeculpence, and I think that
the government wanted to exert more pressure on my side over that fragmentation and that, for what they wrote to
me, people, if they wanted one thing, it would be something that I'd talked about, every now and then, did not
think too much would go into need for
information as the policymaker in this problem, with that said, and that information is now being written, without
any further words, giving it certain Future for publication, for the public purpose.  We time of the day is in
actually gathering information, so, may people enthusiastically made the case for uses of granular information
like that to be agencies.
ROLNEY:, when you talked to F.B. O'Neill earlier this year, he said that he needed to develop a plan for the
foreign policy until 2009, begin with.  And said that that was a part of the Nation PR strategy, and so, any
attempt at vision of foreign that was going to be adopted during that time, and obviously the 2009 for approvals,
never got made.
HATTE: And when was it going to be whatever was being said that we would assemble there in the public information
substantial statements from the advisors, and when the advisor said,, look, I'm sure I am look into the details
of this policy.  If had to draw a line and protecting military interests, just in order to bring us to a war
authorization, right, then I do not think we should have heard anything when it said something like this.  And so,
I really have understand that the policy had wrong, that it had nothing to do with the Cyclone or the Coles attack
whatsoever.
My point to my advisor is nobody, he years old or younger have ever followed any of thoses and correspondence, is
so different in understanding what went wrong or right.  Right prior to known events, Tuesdayine, NATO, in terms
referred to in the US the former Yugoslavia and Afghanistan had been attacked.
NATO attacks Iraq of that kind, and with a field policy document written throughout the brought together people
on the White House and the others within the program, and you have a grasp of this, you start to understand that
they were influencing entire the policy, and he was actually an intermediary in the too White House, I know he had
an honorable appearance here about a minute before the attacks as well, and gave this original perspective on his
decision.  The letter to the Pentagon in my revision said that it had been written, and that, in his testimony, he
had
had been writing for at least two years, and so it was a first.
And so I went over a little bit of the transcript, accompanied by Defence Secretary Gates and was able to go
around the thought and reaction from those in the White department, and when I seen it had happened, and something
on the subject that had been there, I am to classified that having cut off my communications communications to
see the narrative, but also that the allegations were dropped and that there was no sitting in the Beltway, I know
that there were very much re-classified papers and text out documents.
Today is the first of mine to produce a grade in an whopper in a professor of sociology and the megalopolis
department at the Institute of Technology.
The "do
about the nuclear weapons, a regime with a new politeall to the national media account,about,How do we prevent
it from the current situation it isat in France, how many questions are the U.S. on France subject or subject
to, and then with regard to the effects, how do the reports written and distributed get to the Washington.  We're
trying, of course, please as you some material relating to things like corporate intelligence, is going to be
important in number mind, And so because of the secrecy at the State Department I wanted the story probably to
be corrected, and I was going to break the law.  But some of the information found there, at in that time of day,
the day after the attacks, and then also Russian spreading text over the whole internet, and then to misusethe
intelligence on,is caught in the surveillance itself,, and they completed throughdisrepublic of information, of
any military,<|endoftext|>
```

*Figure 9.* Unconditional OpenWebText samples from SDTT with 8-step decoding.

| Di4C samples | Sampling Steps: 4 |
|---|---|

<|endoftext|>.  for the group violence East Israel in Hebrew students.<|endoftext|>Looking for news you can trust? Subscribe to our free newsletters.
Shortly before15 p.m.  an August 12-year- terror suspect in North Carolina, Chicago on his actions were passive line when police officer who appeared to be going to the Milford police area was his family, or even one actually got into a law enforcement checkpoint in a police center, around 30 miles west of,, considering a gun assault.
Read more
That left the NSA's ticket apparatus and state family reward police with much more AP reports, one of the big news stories, the scandal at Cambridge was Clinton's last
because of lack of action questions, so, U.S. government suggested that backlogs actually exist to determine the cause selected several deadly incidents like the, the, and the seizure of of weapons of mass destruction weapons, in, in 2008, was executed in, And beyond.  In the case of theblower in Berlin, near the end, the man was called " part of instead of " "a whole."
Take the NSA's system 30 days later
If the coerced interrogation is true and EL the 1961F's's conduct have nothing to show for the which information that have been made
available, let alone have draw any conclusions.  There is not a reason to believe they noticed the man in Berlin ho their donut in, hoping to pass their prison facility and treat the press cell- in a also that evening
He weighs Jr.  might weigh 40 pounds or was inside the man, armor.  exceeded the limit We said to on the newer L1 and left too late, but so sophisticated hadn't ever drawS. activity, and so we extended compensate for flights along that route here and there, leaveiliated.  mega.  media groups done as this
play out would and taken into consideration.  several calculations.  So here is cyber penetration, degraded aircraft being raised, issues that the United get soon and extremely sensitive.
So on the other hand, the beautiful question, why we decided to release the answer 16, retain not only the existing standards but
20 to 30 more pilots in well.  This is why this has announced this troop mission, and the demonstration their us Come to meet those more.
The initial tests that's haven't been released have been reported to us
as abstractions, but for one of their major formal build up ahead of a very 2013 sales, Mi Type 16 is really beating the standard that they are now or, at least, being (almost) ready final forward soon, we posted some tests for the pilots.  Mi3520 is now available, was afince.be submitted, and had begun the substantial formal road of arguing with the Inc.  team with the military over the current.  This
nilly is a orMPV for all reluctant pilots, but is out of sight, which will allow the APV to run up to 3 months, so that is time for final submission., they will still be running for weekend suicide test or several months they submit to GM.
But we know Harri and I have are with to on version robot, and are happy with this test score.  We
are told.  that those trained will work with a better close combat.
After Iraq, air fight is a lot of flagrant affairs for a relatively non-military air force, and with their performance, Gen 2 can paint a in the galaxy , looking more natural.
So, not know sure is what I am next for, this should alive
evenis being given very early, but chances are there is the ' Aircraftless CBA Lecture.'  is a schedule new,which my friends call Trabi..PLA, who possesses this skill, but this skill is just balanced, rolled that getting the A of time to adjust to the battlefield.  I. MUDoldt coming in with me, which also helps, though a month or so Trauma announcement wasally be possible.  Only before in case of a major to report on, I have brought the power, other than driveBN within NeverLsit,ying something features like his list of old NA 10 is he is a good,
ie some people may not have understood and conducted as well as I said, but since 2 years on, most players even on the PC
have now thought I brought in general much better alignment, and that that's good
a the brought the wide field ability, and he
really reached against the Donald come in and put ground game together that didn like you can take advantage of type aerial.  He looked like a good guy that really won
you so many of these might consider and may for normally guy's perspective , great brought", and it went on from win to loss,
though many surprises these<|endoftext|>

*Figure 10.* Unconditional OpenWebText samples from Di4C with 4-step decoding.

| Di4C samples | Sampling Steps: 8 |
|---|---|

<|endoftext|>, moved into Europe.  I was just trying to describe how different it was as a player and this six months really I was working with TeamLiquid for the first time of a brand new career.  And I had an experience the been.  I actually spent a lot of time back in Europe, so I did acknowledge during that period a good and full early period as to start to see the correlation between where I was been and where I going.  But I was definitely noticed that every time that I moved in, there was an opportunity for me to better the floor of.  It became a, well I can't help but create an image of yourself, that you was known for your true professionalism and least work hard work.  From some of my looks at people.  And I was praised that year for putting Team Liquid against very good teams and then waiting much later in North America, July to have a good staff work together on the team and that was the I'm hoping to use to better our floor of, because that is one that I Love most of, has to be one of the most memorable positions I ever ever held at Pro Circuit and one of the things that has soaked up my, a fair amount of time in the pro circuit, the experience and knowledge, a core of this applies to the Pro Circuit tournament that will usher in its return.  Gul says that beginning in 2017 the Internet, most newspapers and television were in a crash course where they the rules for one whole game, and that in the case of, World Champion No.  2 Tomas Rosas Corbin earlier this month, put pilot devices into the mechanics, and seen in an exhibition by some other people and games, which clearly indicated that we were not to do anything that do."Meagan encouraged me not fall into this trap," him continues, "because, 'Why you become stuck on thing after thing, you can't answer'. I'm say, this tournament has nothing really new beyond that to look at, but one major change, which has been, is a major milestone.  This means that several weeks, given that this game has changed dramatically, there is some big ball selection and stability improvements from from 2016, when a couple of testing events were held around which structures and tracks."This Test will run largely on a fixed minerals, standard structure that we had originally designed," Kiswani says, incorporating that again will be through reviewing the pilot program.  Additionally, one of the most important aspects of the Test will be a lot of the match-encoding, that are bringing almost the match on track, and removing and eliminating the weird interconnections between the players involved.  Technically, major tournaments will not held in the postseason for in changes, the most being 1000s USL events; being held in the E Tempo E, and, the Openrace event; and continuing in the Openrace Series, at some non-crowd games.  The Test will run once a month.  After that."
This will continue to the Pro Circuit with another ten hours available, he says.  He does not anticipate these changes this time, unfold in 2017, and that we will need to determine trying some of TeamLiquid's tweaks to combine with this shift from their current status quo, so it will take more of a concerted for so many of these tournaments to continue in 2017, for example, this autumn.The fifthplace Challenger Series, which continues on, has no impact on the pros involved, or, by the way, players that had waited until now to participate there, but is the only tournament where a player had a strong season prior to the filing of Challenger,.  In addition, there is very significant, very different, and very slow structure of the game that don't cater to the players," Kiswani explained in an interview "Last year, at the World Championships Lirotti Petrova, in theBecause of the FIFA clock the event is slowing down significantly.  The extensive testing and Harrisley to was significantly downsized in the, and Natus, for this tournament as well, and it was the first tournament to introduce a player to DTSR and the testing introduced to also brought some potential improvements to the previous players system.  However, its effectiveness has been cast in doubt by, who do the marketing players and management teams summit together humans, which not only does not reward the best people with action, but also provide the best hard working professionals that to examine their ability to compete; but it also give for them to, finally, just take some time to prepare for it.Now, of some, the fact of that is a little bit, I've been very happy with the progress is are making at Pro year.  For every single event playing played, it represents a less balanced, and not as big, same shuttle pool. This competition is to check how to cater to the player needs, and, as we talked about from the start there is no<|endoftext|>

*Figure 11.* Unconditional OpenWebText samples from Di4C with 8-step decoding.

**IMDM samples**                                                           Sampling Steps: 4

```
<|endoftext|> the confidence that we had enjoyed in our business, so there was also a clear tone of information
culture and respect.  Holm has been planning processes for the a year, like our healthcare transition, the
certification process.
"We will continue to observe the course of the BAE negotiations on the aim of improving and developing environment
conditions on the open market that will close the gap between time and money.  The development process.  It is
very straightforward:  of course, the majority growing proud of the game will come to to it from other countries
and other regions, once you were able to purchase the game.  The LAMB staff who are in working alongside 30
minutes of recent layoffs in terms of resources, have been in talks to find were also recently at IGN.
Manens Singh is a specialist in information culture research who has currently negotiations with Sony for the
new parent company, Sony.<|endoftext|>Holm Singh acknowledged in an interview that he had seen the transformation
in the political philosophy to one that is now associated with traditional British institutions and sought to
use victims of the left to behead populism.He said that, Holm is feeling something of this surprising team about
the very complex climate that is sweeping Britain." This yearidd has had a stern along front row with two League
Media de Mrove ministers, who suggested that the government should break up the Conservative following Keith David
Cameron.  Indeed, that suggested that the British education community disagreed with the decision Comment the
voters in Boxton on the realities in the UK. Moreover week he also responded to the statement from the teachers
association, and that the work have to have some historical context with Trump.  breaking up the Conservative
party has intended to respond to the highly controversial, the first time in his presidency, following the shocker
election results that have seen angry verdicts, led to engagement, of 'Stop' you death campaigner.  "our current
flaw is what we think is a nasty habit of saying a person who failed will 'stop you off' the job."idd has taken a
separate reaction from the Metropolitan Department and the Mayor's representatives, who believes that he is the
claiming a majority of some of the resignations.  "My view Murdoch was that it does not necessarily seem there
have been any demands of response of their concerns administrators at the Met.  And with withIt needs to show
they are going to become partB the a force," he told the committee, the Director of Crime and Crime, the Cabinet
spokesman Committee on Crime and Strategy, who was committed toOn the basis of changes, those changes should be
positive across the working group."
Read more:
On Tuesday Jan 26 9, 2010 theDispatch reported that federal officers were on video for an extended period of time
over the past two months.  And Wednesday, Jan.  16, 2010, Wednesday evening, the Wall Street Journal reported that
the Met became the first law enforcement agency to gather evidence regarding the deaths of officers.
Police spokesman James Cole, 3:06 p.m.  on Justice matter officials reported that has been completed.  However,
daily theNews the Times reported that the matter speaks to a Post Director's court.  Cole emphasized this is
about the proceed of discovery, not media speculation.  The day beforeazily filed a public request to comment on
television and there was NO reply of a statement involving the matter reported.  InsteadAshley Jarrett said both
the Met and Metropolitan police would be reviewing the issue of that public, though details of how long that will
remain in the investigation.
"First, the (is) investigating commissioner (SOC) actions were met with a resolution that was designed to direct a
permanent change to the data security issue, Jarrett said, adding Metropolitan Police is investigate what was
done.  Second, it intends that it will respond to the Times on morning after understanding the full details
of the matter, McCullke said.There is a settlement reached from the Met and FedEx Corporation in Seward after
the Times published the findings.According to spokesman Edward M Lynch said that Lynch told the Daily one day
after the video published that the company was also investigating, and that It was contacted last year by a
Justice Department inspector, which assured stated that she was determining that no evidence, including that
an internal affairs division had had recovered all three evidence and that the in-the coverup video had been,
because they were warned that they posed the greatest threat to public safety.  It was also asked what evidence it
had by<|endoftext|>Connection were hacked and company was not FBI Despite government green light.  (The has a 2012
government contributing decision) From a m Feds.
The government created an Internet page that led to a Drew Stafford lawsuit over "SdnDoT.com." The story on
the website allowed any of the evidence to be brought into federal court without the need of a warrant for
charges.
This evidence was relevant in a second open hearingWC case as of, July 3rd 2012:.
raar the delays<|endoftext|>
```

*Figure 12.* Unconditional OpenWebText samples from IMDM with 4-step decoding.

**IMDM samples** — Sampling Steps: 8

```
<|endoftext|> determine if a is favorable to abortion."
He acknowledged that working with Planned Parenthood is a "serious matter, though" and the most important thing
is that the company doesn't have a political agenda.  It aims to serve people, as was the case of T-Mobile years
ago, not as big was business with them.  Trump, the CEO emphasized, is important on Planned Parenthood, and he
did build a great deal in his relationship with the group, which he said, "What is is relationship and what is
dependent on that,"
International players responding to its
I delivered multiple news about its decision to the world players on Twitter.  Early this month I tweeted the
same Pinterest the court ruled on Monday in favor of the United States, we planned to attend the G3 simultaneously
in July.  Although we are still in shock with the court decision, we would like to say thanks to our supporters
and everybody, for working to make these releases available for free!  @ attack4wquality<|endoftext|>Sony's Greg
Jaffe has the been working on phone and video interactions for weeks, and is confident that these issues will be
resolved over the coming weeks.  This is not necessarily going to be a public report, and as Jaffe explained in
a Monday statement in which the feedback from many in industry agrees that in many ways is a lot we are looking
to see.  "The channel works themselves out, and there's a lot of fantastic feedback from the gaming community,
and we want to take that from our point of view making these a more engaging games," Sony said in the statement.
Sony has also been looking into a number of other entertainment libraries, however it's more than just the to-game
multiplayer experiences.  Activision actually put together some impressive titles, with the release of Thanos
and Command, games that sold over 775 million copies.  That echoed our sentiments that we think this is a very
attractive way to reach the masses, with even Kevin Spencer, senior VP of visual at Xbox, noting that funding is
low and were excited to make this stay in place company when other resources are constrained.
It's too early to say anything concrete about the details of the new multiplayer as many're still trying to figure
out, but in May, when Activision announced Hel for Link, I'd told IGN that Activision's goal is to ship the game
on both One and PC, and that will do all of the other work that is needed on the game after installation.  A
multiplayer that will be online, add new characters and colors, that can kind of play on the fly, and will sort
of follow the focused, real-world elements of the game.  We've found to be lots of fun pieces of Hel that can
resist used unnecessarily, and the library we work with will be a mix of everything from player interaction
features to animation.  As for result, it's unlikely to come as a surprise that an our first goal will be to
make Hel as a more interactive experience, which is also certain to be central to the game as well, as well.
Hel will in turn help setting interaction up for talented performers to share things they've wanted to work
with, share, etc.  This was a priority because there was so many feedback in the community, that we started
taking steps to address that and made it easier for us to add features like that.  We've sort of increased up
the game, approaching things character and in the story, and I'm sure there is some out there where there's fewer
characters.  But I think that we're really enjoying the enhancements to Android that people in the industry are
compelled to forward to seeing.  Hel is really helping to make sure we make the game a great entertainment library
for people, and built the game in a way that binds it together now that things have been assessed.  We've been
working guidance with a lot of teams on the game interface and overall tech and experiences, how devices work with
third-party conversion functionality and the overall tech and experiences that the communication, all of the UI
elements we're creating and managing those transitions on landscape.  And with the kind of rendering capability
Sony has, the game interface will now have the ability and even ability to the on-in feature set automatically
play in mode in as little as seconds.  As for each visual piece that comes to the game, the piece will be open
sourced almost immediately after its output to the Xbox software, and they're also doing special effects, as it's
clear they wanted to work on some planned shots to see what effects they can were up.  This means, as long as
they already figure out, they'll want to include some special effects when delivered to the PC. "Having been of
Sony's month left me unimpressed that the game feels more...the last one Sony made almost eight years ago," the
somewhat<|endoftext|>
```

*Figure 13.* Unconditional OpenWebText samples from IMDM with 8-step decoding.

