# OpenReview forum: "Infinite Mask Diffusion for Few-Step Distillation"
_ICML.cc/2026/Conference — ICML 2026 regular_

### Official Review · Reviewer_CLRN · 2026-03-02

**Soundness:** 2
**Presentation:** 3
**Significance:** 2
**Originality:** 3
**Overall Recommendation:** 3
**Confidence:** 3

**Summary:**

This work analyzes the lower bound of the factorization error in MDLM sampling, points out that the usage of single-state mask in MDLM leads to an irreducible error bound, and proposes an infinite-state mask diffusion model (IMDM) to mitigate this issue. IMDM could leverage the pretrained weights of MDLM for efficient adaptation. With a theoretically zero-error optimal model existing, the distillation and few-step generation performance could be improved.

**Compliance With Llm Reviewing Policy:**

Affirmed.

**Key Questions For Authors:**

Please see the weaknesses section.

**Limitations:**

Please see the weaknesses section.

**Strengths And Weaknesses:**

Strengths

- This work theoretically analyzes the lower bound of factorization error in the MDLM parallel decoding process from the mutual information perspective, which locates the source of MDM sampling error precisely.
- This work proposes the infinite mask diffusion model (IMDM), which theoretically guarantees the existence of an optimal model with zero factorization error. IMDM’s implementation is simple and can be adapted from pretrained MDMs.
- Experiments show that IMDM achieves lower generative PPL and higher MAUVE compared with other distillation methods.

Weaknesses

- There is already a similar work analyzing the sampling error of DLM from the perspective of mutual information [1]. Besides, [2] is similarly motivated by modeling the token correlations to reduce the sampling error, and [3] also uses a Gaussian-based soft mask to surpass the single-state mask in MDM. Could you comment on the novelty/advantage of IMDM over [1, 2, 3]?
- The reason why simply adding stochasticity to masks or adding more mask states can reduce the sampling error of MDM is hard to understand. The mechanism of error reduction is not fully explained. Although the minimum error is 0, the mean/variance of error compared with MDM is not analyzed. Reducing an error lower bound cannot guarantee a lower error.
- The experiment seems incomplete, this paper only evaluates the generative performance, the evaluation of zero-shot language modeling perplexity is missing.
- The model scale is limited, as IMDM is a lightweight fine-tuning method, it will be much more convincing to fine-tune larger MDLMs, e.g. llada, dream. It's unclear if the reported gains hold or if new instabilities arise at larger scales.

[1] Gen Li and Changxiao Cai. Breaking AR’s sampling bottleneck: Provable acceleration via diffusion language models. In The Thirty-ninth Annual Conference on Neural Information Processing Systems, 2025.

[2] Tianyu Xie, Shuchen Xue, Zijin Feng, Tianyang Hu, Jiacheng Sun, Zhenguo Li, and Cheng Zhang. Variational autoencoding discrete diffusion with enhanced dimensional correlations modeling. In The Fourteenth International Conference on Learning Representations, 2026.

[3] Huangjie Zheng, Shansan Gong, Ruixiang ZHANG, Tianrong Chen, Jiatao Gu, Mingyuan Zhou, Navdeep Jaitly, and Yizhe Zhang. Continuously augmented discrete diffusion model for categorical generative modeling. In The Fourteenth International Conference on Learning Representations, 2026.

---

> ### Author Rebuttal · Authors · 2026-03-31
>
> We thank the reviewer for the constructive feedback.
>
> > **W1:** Novelty over [1, 2, 3].
>
> We appreciate the references. We briefly comment on each.
> Li et al. [1] analyze sampling error theoretically; IMDM proposes a practical method to reduce factorization error via infinite stochastic masks combined with distillation.
> VADD [2] introduces data-dependent stochasticity via a VAE encoder, whereas IMDM's data-independent noise serves as a mask category simulator. VADD requires an additional VAE encoder-decoder which requires from scratch training, while IMDM adds a lightweight MLP, enabling reuse of pretrained weights and existing distillation methods.
> CADD [3] introduces a new framework with continuous Gaussian noise that is not directly compatible with pretrained MDMs or existing distillation methods. IMDM preserves the MDM framework (Eq. 5-6), enabling seamless reuse of both.
>
> ---
>
> > **W2:** Mechanism of error reduction.
>
> We agree that reducing the lower bound alone does not guarantee lower error; however, IMDM achieves lower error in practice. The mechanism involves two components: IMDM provides the capacity for correlated predictions, and distillation exploits it.
>
> First, IMDM creates the capacity. In standard MDMs, every masked position receives the same [MASK] token, so the model has no basis to commit to one coherent outcome and resorts to outputting independent marginals. In IMDM, each masked position receives a distinct random noise, which carries no information about the original data. This shared source of external randomness allows the model to map different noise combinations to different coherent outcomes, allowing all positions to agree on a consistent prediction. This removes the structural barrier (Thm. 4.1, 4.2) that prevents error reduction in MDMs.
>
> Second, distillation methods such as SDTT and ReDi can be interpreted as implicitly minimizing the total correlation (Eq. 7), which drives the model to reduce factorization error through the stochastic masks. Without distillation, there is no training signal to reduce factorization error; with distillation, the noise becomes the channel through which the model learns consistent predictions.
>
> The reduction of error with IMDM is demonstrated in the synthetic experiment (Sec. 5.2), where MDM achieves only 49.8% validity while IMDM achieves 97.7% on the {00, 11} task with 1-step decoding (Table 1). As shown in Table A, MDM outputs $p_\theta($tok$=0) \approx 0.5$ for both positions regardless of $\epsilon$, since all masked positions receive the same [MASK]. In contrast, IMDM produces near-deterministic, internally consistent predictions that vary with $\epsilon$: (0,0) for $\epsilon_A$ and (1,1) for $\epsilon_B$.
>
> We additionally measured $TC(z_0 | z_1)$ = $D_{KL}(p_{data}‖q_{model})$ directly, where $q_{model}$ is the factorized joint under full masking (IMDM marginalizes over 10,000 random $\epsilon$). MDM achieves $TC=0.693(=\ln 2)$, while IMDM achieves $TC = 0.081$, an 88% reduction. This confirms that the stochastic masks reduce factorization error in practice, not merely its lower bound. On real data, the quantity is intractable as it requires p_data. However, the consistent improvements in Gen. PPL and MAUVE serve as indirect evidence.
>
> **Table A. Per-token output probabilities in the synthetic task.**
> |Model|$\epsilon$|$p_\theta($tok1$=0)$|$p_\theta($tok2$=0)$|
> |---|---|---|---|
> |MDM|A|49.7%|49.7%|
> |MDM|B|49.7%|49.7%|
> |IMDM|A|98.2%|100%|
> |IMDM|B|0.6%|0.4%|
>
>
> ---
>
> > **W3:** Zero-shot language modeling perplexity missing.
>
> We finetuned pretrained MDLM to IMDM on OpenWebText for 10k iterations and evaluated zero-shot perplexity following [4]. As shown in Table B, IMDM matches MDLM across all datasets, confirming seamless transition from pretrained MDM weights.
>
> **Table B. Zero-shot PPL.**
> ||Wikitext|LM1B|Lambada|AGNews|Pubmed|Arxiv|
> |---|---|---|---|---|---|---|
> |MDLM-ft|32.12|65.21|46.68|60.48|43.30|37.80|
> |IMDM-ft|32.17|65.36|46.80|60.56|43.36|37.87|
>
> ---
>
> > **W4:** The model scale is limited.
>
> IMDM scales effectively to 1.1B parameters. We finetuned the pretrained SMDM 1.1B model [5] as IMDM with SDTT+ReDi on SlimPajama, and results in Table C show consistent improvements. Gen. PPL is evaluated using LLaMA 3.1-8B (seq. len. 2048), so absolute values differ from OpenWebText experiments.
>
> **Table C. Gen.PPL of SMDM on SlimPajama.**
> |Steps|SMDM|IMDM|
> |---|---|---|
> |2|3102.37|2436.04|
> |8|1717.38|847.28|
> |32|728.28|245.80|
> |64|436.10|177.13|
>
> [1] Li et al. Breaking AR’s sampling bottleneck: Provable acceleration via diffusion language models. 2025 \
> [2] Xie et al. Variational autoencoding discrete diffusion with enhanced dimensional correlations modeling. 2026 \
> [3] Zheng et al. Continuously augmented discrete diffusion model for categorical generative modeling. 2026 \
> [4] Arriola et al. Block Diffusion: Interpolating Between Autoregressive and Diffusion Language Models. 2025 \
> [5] Nie et al. Scaling up Masked Diffusion Models on Text. 2024

---

> > ### Author Rebuttal · Reviewer_CLRN · 2026-04-01
> >
> > Thanks for your response.
> >
> > W1: addressed.
> >
> > W2: this major concern is not fully resolved. "External randomness allows the model to map different noise combinations to different coherent outcomes", which is the core of this paper, so a formal theoretical analysis is necessary.
> >
> > W3: addressed.
> >
> > W4: not fully resolved, the Gen. PPL should be reported with the unigram entropy. [1]
> >
> > [1] Kaiwen Zheng, Yongxin Chen, Hanzi Mao, Ming-Yu Liu, Jun Zhu, and Qinsheng Zhang. Masked diffusion models are secretly time-agnostic masked models and exploit inaccurate categorical sampling. In The Thirteenth International Conference on Learning Representations, 2025. URL https://openreview.net/forum?id=CTC7CmirNr.

---

> > > ### Author Response · Authors · 2026-04-02
> > >
> > > We thank the reviewer for the thoughtful follow-up. We are pleased that W1 and W3 have been resolved, and we provide additional clarification on the remaining concerns regarding W2 and W4 below.
> > >
> > > > **W2:** Mechanism of error reduction (continued)
> > >
> > > We appreciate the reviewer's emphasis on this point. We recognize that our previous rebuttal presented the argument informally, and we would like to clarify the mechanism identified in the constructive proof of Theorem 4.2 (Appx. A.2), how the quoted statement connects to it, and how the toy experiment supports it.
> > >
> > > The constructive proof of Theorem 4.2 (Appx. A.2) achieves zero factorization error through a partition-and-map mechanism. The proof considers the set of positions $E$ that are masked at timestep $t$ and decoded between timestep $t$ and $s$. For these positions, given the unmasked context (the tokens outside $E$), the proof constructs a deterministic map $F$ that takes the noise ($z_t^E \in \mathcal{M}^{|E|}$, where $\mathcal{M}$ is IMDM's infinite set of mask states) as input and outputs a token sequence that could be sampled from the true conditional distribution $p(z_s^E | z_t, e_E)$. Because $|\mathcal{M}| \to \infty$, the noise $z_t^E$ can take on infinitely many distinct values, providing enough diversity for $F$ to partition them into subsets with sizes proportional to the target probabilities, each assigned to a different valid token sequence. The model deterministically outputs each sequence based on the noise, and since marginalizing over the noise recovers the true conditional distribution, this yields $TC_{\theta^*} = 0$.
> > >
> > > Each element of the reviewer's quoted claim, "external randomness allows the model to map different noise combinations to different coherent outcomes," directly corresponds to a specific object in this construction. "External randomness" refers to the noise $z_t^E \in \mathcal{M}^{|E|}$. "Map" refers to the deterministic map $F$. "Different noise combinations" correspond to different inputs to $F$ (i.e., different values of $z_t^E$), and "different coherent outcomes" correspond to different outputs of $F$ (i.e., $F(z_t^E) \in \mathcal{V}^{|E|}$). In MDMs, this construction is not available: all masked positions receive the same deterministic mask token due to the lack of "external randomness," so $z_t^E$ takes a single fixed value. With no variation in $z_t^E$, there is no set to partition, and the model is restricted to outputting independent marginals regardless of its parameters (Theorem 4.1). In summary, the quoted claim is equivalent to the following formal statement: the deterministic map $F$ partitions $z_t^E \in \mathcal{M}^{|E|}$ so that its pushforward reproduces $p(z_s^E | z_t, e_E)$, which is precisely what the proof of Theorem 4.2 constructs.
> > >
> > > The synthetic $\{00, 11\}$ task (Sec. 5.2) provides supporting evidence that the learned IMDM recovers a solution consistent with the above construction. As reported in Table A of our previous rebuttal, the learned model produces almost deterministic, coherent predictions that vary with the noise realization $\epsilon$: under $\epsilon_A$, both tokens are predicted as 0 with over 98% probability (98.2% and 100%), and under $\epsilon_B$, both tokens are predicted as 1 with over 99% probability (99.4% and 99.6%). That is, different noise realizations are routed to different coherent outcomes, $(0,0)$ or $(1,1)$, mirroring the partition-and-map structure described above. At the same time, as shown in Table 1, the overall token entropy (0.69) matches the ground truth and the validity reaches 97.7%, indicating that marginalizing over the noise realizations recovers the target distribution. In contrast, MDM outputs near-uniform predictions (49.7%) for both tokens regardless of the input, consistent with the absence of a partitionable set of mask states. While MDM also matches the token entropy, it fails to capture the inter-token dependency, resulting in validity comparable to random sampling (49.8%).
> > >
> > > We thank the reviewer for encouraging us to articulate this more clearly. In the revision, we plan to bring the key intuition from the constructive proof (the partition-and-map mechanism and its connection to the entropy of the masked state) into the main text (after Theorem 4.2), along with an explicit discussion of how the synthetic experiment validates this formal construction (Sec. 5.2).
> > >
> > > > **W4:** Gen. PPL should be reported with the unigram entropy
> > >
> > > We report the unigram entropy corresponding to Table C in Table D below (real data: 5.76). IMDM's entropy is consistently closer to real data than SMDM across all step counts, confirming that the Gen. PPL improvements in Table C reflect enhanced generation quality.
> > >
> > > Table D. Unigram entropy.
> > >
> > > |Steps|SMDM|IMDM|
> > > |-|-|-|
> > > |2|6.36|6.27|
> > > |8|6.33|6.20|
> > > |32|6.29|6.13|
> > > |64|6.25|6.08|
> > >
> > > We hope our responses have addressed the reviewer's remaining concerns. We will incorporate all discussed changes in the revision.

---

### Official Review · Reviewer_E2zZ · 2026-03-10

**Soundness:** 2
**Presentation:** 3
**Significance:** 3
**Originality:** 2
**Overall Recommendation:** 4
**Confidence:** 5

**Summary:**

This paper proposes changing the source (reference) distribution in discrete diffusion from an all masked sequence, to a sequences composed by a subset of a vocabulary of uncountably many different masks. It points out that for such a choice there exists a theoretical optimal model where the target is unambiguous, and that the beneficial properties of masked diffusion are maintained. The proposed method is tested in different settings.

**Compliance With Llm Reviewing Policy:**

Affirmed.

**Final Justification:**

The paper's novelty and differences from existing literature have been clarified. The new results add more weight to the significance of the paper.

**Key Questions For Authors:**

**Q1.** Can the authors compare how their parametrization of their infinite mask distribution embedding choice compares to the multimask one? That is, compare multimasked diffusion with its limiting version.

**Q2.** Can authors double check their code?

**Limitations:**

Limitations are not clearly grouped and stated in a specific section.

**Strengths And Weaknesses:**

**Strengths**

**S1**. The writing is quite good, and the paper tackles an important issue.

**S2**. The idea is good, and intuitive. In particular, the insight that in the proposed method there will be an optimal coupling, and that the nice properties of the masked diffusion as well as the loss are maintained.

**S3**. Experiments are quite rich and they use a proper set of metrics: generative perplexity for quality, entropy to check in-sentence mode collapse, MAUVE to check mode collapse.

**Weaknesses**

**W1**. The main weakness is the novelty of the contribution. Prior work [1] defines multi-masked flows, derives the loss which is identical to the one in the paper (a straightforward generalization of masked flows), and point out that the multimasked flows keep the nice properties of the masked flows. In addition, their motivation comes precisely from optimal transport, as they required multi states in order to enable couplings. They even point out that, for a large number of masks, one can approximate any data distribution using deterministic couplings. As such, most of this work's contributions are a subset of one of the main three contributions of that paper. The authors should remove novelty claims and cite [1] about the loss (bound), maintenance of properties, and the optimal coupling (a modified version of the latter can be kept by pointing out that the multimasked flows hint at the optimality in the limit but do not make a formal statement). The limiting part as well as combinations with distillation are completely novel to my knowledge however.

**W2**. There seems to be some issues with the generative perplexity results in the MDM case. From my own experience, and other papers e.g. [2] generative perplexity is considerably lower than what is reported in Figure 3 (for 32 steps for example in their Figure 8). Can the authors please double check their code? In addition, if I am not mistaken,  $p(e_{ij})$ should be defined as the probability that both positions are unmasked somewhere before $s$ and after $t$. The event "simultaneously decoded (unmasked) at timestep s" would have probability 0, in the true theoretical process.

**W3**. Due to self-distillation, entropy drops significantly. The paper does not present results of MDM and IDMD trained from scratch and compare between the two.

**W4**. I would have appreciated if the paper reported unconditional MAUVE results as well.



[1] Haxholli et al. Minibatch Optimal Transport and Perplexity Bound Estimation in Discrete Flow Matching. ICLR 2026 submission  Openreview.

[2] Shi et al. Simplified and Generalized Masked Diffusion for Discrete Data. Neurips 2024.

---

> ### Author Rebuttal · Authors · 2026-03-31
>
> We thank the reviewer for the constructive feedback. We will incorporate the suggested improvements in the revised version.
>
> > **W1, Q1:** Novelty and relationship to Haxholli et al. [1]
>
> We agree that [1] shares the insight of using multiple mask states and that properties of masked flows are preserved under multiple masks. We will revise the paper to cite [1] for the multi-mask formulation and this observation, and will remove any claim of novelty on these specific points.
> That said, we note that the reviewer also acknowledged that "the limiting part as well as combinations with distillation are completely novel." We therefore position IMDM's novelty as follows: (1) the $M \to \infty$ limiting formulation derived from uniform discrete diffusion, which provides a distinct theoretical pathway from [1]'s finite multi-mask flow, (2) the formal analysis of the factorization error lower bound (Theorem 4.1) and the existence proof of an optimal zero-error solution (Theorem 4.2), (3) the continuous-noise + lightweight-MLP implementation that enables direct reuse of pretrained MDM weights, and (4) seamless integration with existing distillation methods (SDTT, ReDi). These contributions are absent in [1], which focuses on finite multi-mask tokens with minibatch optimal transport and requires training from scratch with 50,257 additional discrete tokens.
>
> We also compared IMDM with finite multi-mask diffusion (6,144 masks, matched parameters) in Table A. IMDM consistently outperforms the finite variant, confirming the advantage of infinite mask categories over a finite set.
>
> **Table A. Comparison to multi-mask diffusion (Gen.PPL).**
> |Steps|Uniform|Multi-mask|
> |---|---|---|
> |2|165.85|208.72|
> |4|93.44|101.91|
> |8|65.55|66.64|
> |16|52.51|55.05|
>
> ---
>
> > **W2 (1/2), Q2:** Generative perplexity results and code verification.
>
> Thanks for your constructive feedback. We found that we reported the performance of the running model while the standard benchmark uses EMA models. The updated results are in Table B-E. The score of MDLM matches the scores reported in [2], [3]. As discussed in the main paper, IMDM consistently outperforms the baselines in the few-step regime.
>
> **Table B. Gen.PPL on LM1B.**
> |Steps|MDLM+SDTT|IMDM+SDTT|MDLM+ReDi|IMDM+ReDi|MDLM+SDTT+ReDi|IMDM+SDTT+ReDi|
> |---|---|---|---|---|---|---|
> |2|584.22|524.31|530.36|284.62|353.90|165.85|
> |4|227.08|210.05|195.39|151.89|122.56|93.44|
> |8|126.07|122.52|113.14|104.93|72.04|65.55|
> |16|93.29|90.16|85.27|81.57|55.36|52.51|
>
>
> **Table C. Unconditional Gen.PPL on OpenWebText.**
> |Steps|MDLM|SDTT|Di4C|IMDM|
> |---|---|---|---|---|
> |2|3338.21|877.22|747.66|632.41|
> |4|2009.22|339.73|236.27|167.20|
> |8|826.42|112.66|82.00|62.06|
> |16|346.04|57.74|44.12|34.85|
> |32|193.37|40.41|31.38|25.52|
> |64|141.43|33.21|25.80|21.52|
>
>
> **Table D. Conditional Gen.PPL on OpenWebText.**
> |Steps|MDLM|SDTT|Di4C|IMDM|
> |---|---|---|---|---|
> |2|392.12|222.82|209.31|151.41|
> |4|253.02|117.19|101.27|76.57|
> |8|157.39|69.78|61.05|51.93|
> |16|109.59|52.60|46.14|41.99|
> |32|90.09|45.99|41.47|37.28|
> |64|78.00|42.48|38.60|35.60|
>
>
> **Table E. Conditional MAUVE on OpenWebText.**
> |Steps|MDLM|SDTT|Di4C|IMDM|
> |---|---|---|---|---|
> |2|0.06|0.20|0.22|0.73|
> |4|0.37|0.86|0.91|0.95|
> |8|0.86|0.96|0.97|0.96|
> |16|0.94|0.98|0.98|0.98|
> |32|0.96|0.97|0.97|0.97|
> |64|0.97|0.97|0.98|0.97|
>
> ---
>
> > **W2 (2/2):** Definition of $e_{ij}$:
>
> The reviewer is correct; $e_{ij}$ should mean both tokens are masked at $t$ and unmasked by $s$ within a single reverse step. The proof in Appendix A.1 already uses this definition, so results are unaffected. We will revise L169–L170.
>
> ---
>
> > **W3:** From-scratch training comparison.
>
> We trained MDLM and IMDM from scratch on the LM1B dataset. Without distillation, IMDM performs comparably to MDLM trained from scratch on LM1B: Gen.PPL (114.95 vs. 116.37), Entropy (both 4.32), Val.PPL (30.94 vs. 31.69). This confirms that IMDM does not degrade the base model quality and the improvements in few-step generation come from combining IMDM with distillation.
>
> ---
>
> > **W4:** Unconditional MAUVE results.
>
> Following the evaluation protocol of ReMDM [4], we reported unconditional MAUVE on OpenWebText experiments. As shown in Table F, IMDM performs relatively better than the baselines in terms of unconditional MAUVE score.
>
> **Table F. Unconditional MAUVE on OpenWebText.**
> |Steps|MDLM|SDTT|Di4C|IMDM|
> |---|---|---|---|---|
> |2|0.0041|0.0041|0.0041|0.0041|
> |4|0.0041|0.0041|0.0041|0.0042|
> |8|0.0041|0.0043|0.0046|0.0054|
> |16|0.0042|0.0059|0.0080|0.0115|
> |32|0.0047|0.0091|0.0200|0.0231|
> |64|0.0054|0.0198|0.0385|0.0424|
>
>
> [1] Haxholli et al. Minibatch Optimal Transport and Perplexity Bound Estimation in Discrete Flow Matching. 2025 \
> [2] Sahoo et al. The Diffusion Duality. 2025 \
> [3] Pynadath et al. CANDI: Hybrid Discrete-Continuous Diffusion Models. 2025 \
> [4] Wang et al. Remasking Discrete Diffusion Models with Inference-Time Scaling. 2025

---

> > ### Author Rebuttal · Reviewer_E2zZ · 2026-04-02
> >
> > My concerns have been adequately addressed and I will increase my score accordingly.

---

> > > ### Author Response · Authors · 2026-04-03
> > >
> > > We thank Reviewer E2zZ for the careful review and for confirming that the concerns have been resolved.
> > >
> > > We appreciate the reviewer's recognition of our contributions, including the intuitive and well-motivated idea of infinite stochastic masks **(S2)** and a comprehensive evaluation covering quality, diversity, and mode collapse **(S3)**.
> > >
> > > We also appreciate the reviewer acknowledging that our rebuttal resolved the concerns regarding 1) novelty over concurrent multi-mask work **(W1, Q1)**, 2) alignment of the results with published baselines **(W2, Q2)**, 3) from-scratch comparison to MDLM **(W3)**, and 4) unconditional MAUVE results **(W4)**.
> > >
> > > We will ensure that all additional results and discussions from this rebuttal are incorporated into the revised manuscript.

---

### Official Review · Reviewer_N2UU · 2026-03-12

**Soundness:** 3
**Presentation:** 2
**Significance:** 2
**Originality:** 2
**Overall Recommendation:** 4
**Confidence:** 2

**Summary:**

This paper proposes Infinite Mask Diffusion Model, a modification of masked diffusion models aimed at improving few-step generation. The authors first analyze the factorization error arising in discrete diffusion models when the joint posterior over tokens is approximated with a factorized distribution, and theoretically show that MDMs suffer from an irreducible lower bound of this error due to the use of a single deterministic mask token. To address this limitation, the paper introduces IMDM, which replaces the single mask token with an infinite set of stochastic mask states while maintaining the strict separation between masked and data tokens. Concretely, the state space is extended to include latent mask states, and the forward diffusion process transitions tokens to these mask states according to a modified discrete diffusion formulation. In implementation, the infinite masks are implicitly represented by injecting sampled noise into the base mask embedding via an MLP, producing stochastic mask embeddings while remaining compatible with pretrained MDM parameters. The training objective and decoding process largely follow the standard MDM framework, enabling seamless integration with existing few-step distillation techniques. The authors further provide theoretical analysis showing that, under the IMDM formulation, there exists an optimal model that can eliminate the factorization error, suggesting improved capability for modeling token dependencies in few-step generation.

**Compliance With Llm Reviewing Policy:**

Affirmed.

**Final Justification:**

My concerns have been addressed. I would like to keep my score.

**Key Questions For Authors:**

1. What is the dimensionality of the noise vector that is fed into the MLP for generating the stochastic mask embedding? Since the paper only states that  \epsilon is sampled from a uniform distribution, it would be helpful to clarify its exact dimensionality and whether it matches the embedding dimension. Additionally, have the authors investigated whether increasing the dimensionality of the noise vector leads to improved performance or better approximation of the “infinite mask” behavior?

2. Instead of sampling  \epsilon from a fixed distribution, have the authors considered using a learnable noise representation (e.g., parameterized or learned latent variables)? It would be interesting to understand whether a learnable stochastic component could further improve modeling flexibility compared to fixed random noise.

3. How sensitive is the method to the scale or distribution of the injected noise? In particular, the current implementation samples  \epsilon from a uniform distribution in a fixed range. It would be useful to know in what range the noise scale remains robust, and whether the performance degrades when the magnitude of the noise is increased or decreased. A small ablation study on the noise scale could help clarify the stability of the method.

**Limitations:**

The paper does not explicitly discuss the limitations of the proposed approach. For example, it would be helpful to understand potential drawbacks such as  sensitivity to noise injection.

**Strengths And Weaknesses:**

Strengths.
The paper is generally well written and easy to follow. The motivation, background, and methodological description are clearly organized. The paper provides a theoretical analysis of the factorization error in MDMs.

Weaknesses.
First, the empirical improvements appear relatively modest compared to existing approaches. Although the proposed method consistently outperforms several baselines, the margins are not particularly large. Second, the design of IMDM would benefit from a more intuitive explanation.

---

> ### Author Rebuttal · Authors · 2026-03-31
>
> We thank the reviewer for the positive assessment and constructive questions. We address each point below.
>
> > **W1:** Empirical improvements are relatively modest.
>
> While the improvements at higher step counts are incremental, the gains in the few-step regime are substantial. On OpenWebText at 2 steps, IMDM achieves Cond. MAUVE of 0.73 vs. Di4C's 0.22, and Cond. Gen.PPL of 151 vs. 209. On LM1B at 2 steps with SDTT+ReDi, IMDM reduces Gen.PPL by 53% (166 vs. 354). These gains narrow as step count increases, consistent with our analysis that the factorization error bound is most pronounced in the few-step regime.
>
> ---
>
> > **W2:** IMDM design would benefit from a more intuitive explanation.
>
> The core intuition is that IMDM breaks the symmetry that forces MDMs to average over all possible outputs. In standard MDMs, when all positions are masked, every position receives the same [MASK] token, so the model has no basis to commit to one coherent outcome and resorts to outputting independent marginals. In IMDM, each position receives a distinct random noise, which carries no information about the original data but is observable by all other positions through attention. This shared source of external randomness allows the model to map different noise combinations to different coherent outcomes, so that all positions agree on a consistent prediction. We will incorporate this intuitive description in the revised paper.
>
> ---
>
> > **Q1, Q3, L1:** Sensitivity to noise injection.
>
> IMDM is robust to noise design choices. We conducted ablation studies on LM1B with the SDTT+ReDi configuration. As shown in Tables A-C, IMDM consistently outperforms its MDM counterpart across varying noise distributions (Table A), noise dimensions (Table B), and noise scales (Table C). We note that reducing the noise dimension to 256 leads to noticeable degradation at 2-step generation (Table B), suggesting that sufficient dimensionality is needed to simulate diverse mask categories. Beyond this, performance remains stable across all other choices, consistent with $\epsilon$'s role as a mask category simulator. The key requirement is that different $\epsilon$ values produce sufficiently distinct mask embeddings, rather than the particular distribution or scale they are drawn from. We will add an explicit discussion of noise sensitivity in the revision.
>
>
> **Table A. Ablation study on the distribution of noise.**
> |Steps|Uniform|Gaussian|Learnable Gauss|
> |---|---|---|---|
> |2|165.85|167.48|160.02|
> |4|93.44|95.82|93.12|
> |8|65.55|67.98|66.14|
> |16|52.51|56.26|54.56|
>
>
> **Table B. Ablation study on the dimensionality of noise. (Uniform)**
> |Steps|256|768|1536|
> |---|---|---|---|
> |2|294.41|165.85|163.83|
> |4|116.00|93.44|96.47|
> |8|68.74|65.55|69.26|
> |16|54.86|52.51|57.48|
>
>
> **Table C. Ablation study on the scale of noise. (Uniform)**
> |Steps|0.5|1.0|2.0|
> |---|---|---|---|
> |2|179.19|165.85|164.23|
> |4|94.82|93.44|94.60|
> |8|65.87|65.55|66.10|
> |16|52.99|52.51|53.88|
>
> ---
>
> > **Q2:** Learnable noise representation.
>
> IMDM's noise representation is already learnable: $\epsilon$ is transformed through a trainable MLP before being added to the base mask embedding, so the model learns to map random noise to useful mask representations. We additionally experimented with a learnable Gaussian distribution (parameterized $\mu$ and $\sigma$) and reported results in Table A above. The two approaches perform comparably, suggesting that the fixed-distribution design is sufficient and the MLP provides adequate representational flexibility.

---

### Official Review · Reviewer_Gofk · 2026-03-12

**Soundness:** 2
**Presentation:** 3
**Significance:** 3
**Originality:** 2
**Overall Recommendation:** 3
**Confidence:** 4

**Summary:**

The work proposes a novel method of solving the MDMs factorization error problem via using stochastic infinite-state masks. Authors provide theoretical grounding of their method, verify their motivation on synthetic tasks and show superiority of their method in combination with distillation methods.

**Compliance With Llm Reviewing Policy:**

Affirmed.

**Final Justification:**

My overall assessment is Weak Reject, although I see this as a borderline case.

Overall, I think the proposed method is valid, interesting, and technically promising. The paper addresses an important limitation of few-step masked diffusion models, and the core idea seems meaningful. After reading the rebuttal, I also have a much clearer understanding of the method and its intended theoretical motivation.

However, I think the paper requires substantial revision before it is ready. My concern is not that the method is wrong, but that some key parts of the paper are not presented clearly or convincingly enough.

On the theory side, the current presentation needs to be improved. The rebuttal made the method much easier to understand, especially in explaining why the training can lead to the intended behavior in practice. In my view, these explanations are essential and should be added to the paper itself.

On the experimental side, I still think the evaluation should be strengthened, especially around diversity. Right now, it is hard to judge whether the gains come from genuinely better modeling or partly from a narrower output distribution that improves perplexity. A stronger diversity analysis, ideally with comparison points such as real text, would make this trade-off much easier to interpret.

Overall, I think this is a good paper with a meaningful and promising idea, but it still requires major revision in both presentation and experimental validation. That is why I remain on the reject side, though only weakly.

**Key Questions For Authors:**

* My main concern is about training. How does the model learn a stable mapping from random noise to the correct token pair? In the worst case, two very similar noise values may appear in different training cases, but the correct outputs are different. Then the model seems to be asked to map almost the same noise to different answers. Why is this not a problem in practice? This is also why VADD feels different to me: there, the latent comes from (x_0), so the same latent should not map to different token combinations.

* Because of this, I am still not sure why the training should converge. Can you show a training loss curve and compare them with standard mdlm? Can you also show gradient and weight norms for the MLP that turns noise into mask embeddings? This would help clarify whether the model really learns to use the noise, or whether it may simply ignore it.

* I am also confused by the IMDM forward process in Section 4.2. First, the paper introduces a semi-masked and semi-uniform form of noise in the formula above Eq. (5). Then it says that, because there are infinitely many masks, the coefficient before the uniform part goes to zero. I do not think this is wrong, but I do not understand why this extra uniform-noise step is needed at all. Since this is a masked diffusion method, why not define the forward process directly in a masked-only way? To me, the current presentation feels more complicated than needed and may confuse readers.

[1] Song et al. “Ideas in Inference-time Scaling can Benefit Generative Pre-training Algorithms”, 2025

**Strengths And Weaknesses:**

## Strengths

* The paper is clear and well written. The main idea is easy to follow, and the authors explain their contribution in a direct way. Most of the main claims are supported well.

* The theory part is clear. The authors explain why the standard MDLM method has a basic limit, and they show that their infinite-mask method can remove this limit in theory.

* The experiments support the main idea.

## Weaknesses

* The evaluation for unconditional generation is not strong enough. In Figures 2 and 3, the paper mostly shows Gen. PPL. This score alone is not enough, because a model can get a good Gen. PPL while still producing repetitive text. The conditional metrics are also not fully convincing: Cond. Gen. PPL and Cond. MAUVE can still miss some failure cases. It would be better to show the trade-off between diversity and Gen. PPL for different numbers of sampling steps, similar to [1]. A diversity metric computed over the whole sample set would also be more informative than unigram entropy.

* There are no experiments on larger models. Because of this, it is hard to know whether the method still works well in a stronger and more realistic setting.

* A comparison with a close direct competitor is missing. VADD [2] looks much closer. It also solves the problem of factorization error and conditions on VAE latent while you condition on random noise. I see these methods as very similar, so it is important to compare them.

* The practical value of Theorem 4.2 is limited. The theorem says that an optimal solution exists, but it does not show that training can actually find it, that optimization will converge, or that a model with limited size can get close to it.

* The paper needs more ablations and more model analysis. It is still unclear how sensitive the method is to the choice of noise. It is also not clear why the model does not simply ignore the injected noise, or how much fine-tuning is really needed.

* The paper is not fully reproducible from the current description. Some important sampling details are missing or unclear, such as temperature, top-k, and top-p, if they were used. These choices can change perplexity a lot, so without them the reported numbers are harder to trust.

* There are also small presentation issues. Many formulas are not numbered, for example the IMDM forward process above Eq. (5) and the IMDM posterior above Eq. (6).

[1] Meshchaninov et al. “Guided Star-Shaped Masked Diffusion”. 2025

[2] Xie et al. “Variational Autoencoding Discrete Diffusion with Enhanced Dimensional Correlations Modeling”. ICLR 2026

---

> ### Author Rebuttal · Authors · 2026-03-31
>
> We thank the reviewer for the constructive feedback.
>
> > **W1:** Gen. PPL alone is insufficient.
>
> We provide DIV and generative perplexity on OpenWebText in Tables A-B. Following Meshchaninov et al., we vary sampling temperature to examine the diversity-quality trade-off. All baselines are evaluated at temperature 1.0. We additionally report IMDM at temperatures calibrated to match each baseline's DIV. IMDM achieves lower generative perplexity at every matched diversity level, with the advantage most pronounced in the few-step ($\leq$ 8) regime.
>
> **Table A. DIV on OpenWebText.**
> |Steps|SDTT|Di4C|IMDM|IMDM (matching SDTT)|IMDM (matching Di4C)|
> |---|---|---|---|---|---|
> |2|0.38|0.36|0.35|0.38|0.36|
> |4|0.32|0.29|0.26|0.32|0.29|
> |8|0.25|0.23|0.19|0.25|0.23|
> |16|0.21|0.19|0.16|0.21|0.19|
>
> **Table B. Gen.PPL on OpenWebText.**
> |Steps|SDTT|Di4C|IMDM|IMDM (matching SDTT)|IMDM (matching Di4C)|
> |---|---|---|---|---|---|
> |2|877.22|747.66|632.41|750.28|660.06|
> |4|339.73|236.27|167.20|266.25|210.03|
> |8|112.66|82.00|62.06|103.21|81.49|
> |16|57.74|44.12|34.85|56.92|46.96|
>
> > **W2:** No experiments on larger models.
>
> We finetuned the pretrained SMDM 1.1B model, and results show consistent improvements across all step counts. See W4 of Reviewer CLRN for details.
>
> > **W3:** VADD comparison is missing.
>
> As VADD's code is not publicly available, we compare against their reported results (same dataset and model size) in Table C. IMDM consistently outperforms VADD while requiring only a lightweight MLP instead of a full VAE encoder-decoder. See W1 of Reviewer CLRN for a detailed comparison.
>
> **Table C. Comparison to VADD**
> |Steps|VADD|IMDM|
> |---|---|---|
> |16|194.08|34.85|
> |32|124.23|25.52|
> |64|98.82|21.52|
>
> > **W4-5, Q1-2:** Role of noise, training stability, and Theorem 4.2.
>
> As these questions all concern whether IMDM genuinely learns to exploit the injected noise, we address them jointly.
>
> As discussed in Sec. 4.1, MDMs suffer from an irreducible factorization error because all masked positions receive the same deterministic [MASK] token. Since the model predicts each position independently from identical inputs, it can only recover the marginal distribution at each position, blocking the modeling of correlations in the data (Theorem 4.1). IMDM resolves this by giving each masked position a distinct random noise, which carries no information about the original data but is observable by all other positions through attention. This shared source of external randomness allows the model to map different noise combinations to different coherent outcomes, so that factorized predictions become near-deterministic and jointly consistent.
>
> **Q1:** Why conflicting signals are not a problem.
>
> IMDM does not learn a static mapping from noise to data. The model predicts from the entire input $z_t$, which includes both unmasked context and noises. Even if two examples share similar noise at one position, their full inputs differ due to other positions' noise values and unmasked tokens, so no conflicting learning signal arises.
>
> **W5(1/2), Q2:** Training converges and noise is actively used.
>
> We report from scratch training dynamics in Table D. The losses converge comparably, confirming training stability. The MLP weight norm grows steadily (65 → 108 → 256), indicating the model increasingly relies on the noise pathway. We also verified through noise ablations (see Q1 of Reviewer N2UU) that reducing the noise dimension from 768 to 256 degrades 2-step Gen. PPL from 166 to 294, directly confirming that the model depends on noise expressiveness. Additionally, on the {00, 11} synthetic task, we confirm that predictions shift coherently with noise and factorization error decreases by 88% (Table 1; see W2 of Reviewer CLRN for the full analysis).
>
> **Table D. Training Dynamics of MDLM and IMDM.**
> ||100k|200k|500k|
> |---|---|---|---|
> |MDLM-loss|3.81|3.75|3.60|
> |IMDM-loss|3.84|3.76|3.64|
> |MLP-weight norm|65|108|256|
>
> **W4:** Role of Theorem 4.2.
>
> We acknowledge that Theorem 4.2 guarantees existence, not convergence. Its role is to show that IMDM removes the structural barrier of Theorem 4.1, while distillation serves as the practical optimization mechanism.
>
> We are happy to discuss any of these points further if the reviewer has additional questions.
>
> > **W5(2/2):** Fine-tuning.
>
> IMDM requires no separate fine-tuning; the distillation process itself is the only training beyond the pretrained checkpoint.
>
> > **Q3:** Mask-only definition of IMDM.
>
> Defining the forward process directly in a masked-only way is possible, and the final IMDM forward process (Eq. 5) is indeed purely masked. The semi-uniform form above Eq. 5 exists only as an intermediate derivation step, not as part of the actual model. We chose this path to directly reuse the NELBO for uniform discrete diffusion [1] and take the $M \to \infty$ limit (Appendix A.3). We will clarify this motivation in the revision.
>
> [1] Schiff et al. Simple Guidance Mechanisms for Discrete Diffusion Models, 2024

---

> > ### Author Rebuttal · Reviewer_Gofk · 2026-04-04
> >
> > Thank you for the detailed rebuttal. It helped clarify several points, but a few important concerns still remain for me.
> >
> > First, my main concern is still about training rather than existence. The paper and the rebuttal show that a good solution may exist in principle, but they still do not explain why standard training in a finite model should reliably find such a solution.
> >
> > Second, because of this, Q1 still does not feel fully resolved to me. The hard case is when the context stays the same but the sampled noise changes. In that setting, the model still has to divide a continuous noise space into regions corresponding to different valid outputs. The rebuttal points out that the full inputs are different, but that does not really address the core issue. The main question is why this partition of the noise space should be stable and learnable in practice, rather than arbitrary or brittle.
> >
> > Third, I do not think the rebuttal is fully consistent with the paper’s own theoretical argument. My concern is not that Appendix A.2 constructs a deterministic mapping from noise to outputs — that is in fact how the paper establishes the existence result. The issue is that the rebuttal then says IMDM does not learn such a mapping. To me, this leaves the practical question unanswered: if the method is motivated by the existence of a good noise partition, why should training actually discover it?
> >
> > This matters even more because IMDM is initialized to behave exactly like the pretrained MDM, since the output layer of the noise MLP is zero-initialized. So the practical question is not only whether a better noise-dependent solution exists, but also how training reliably moves the model away from the solution that simply ignores noise.
> >
> > I am also concerned by the diversity trend. In the paper, unconditional entropy on OpenWebText steadily drops as the number of steps increases. In the rebuttal, DIV also decreases with more steps. This seems unusual and deserves more explanation. It raises the possibility that some of the Gen.PPL improvement may come from a narrower output distribution, rather than only from better modeling.
> >
> > There are also some inconsistencies in the new quantitative results. The new IMDM OpenWebText numbers reported in the rebuttal do not match Table 5 in the paper. Please clarify whether these results come from a different checkpoint, a different temperature, or a different evaluation setup.

---

> > > ### Author Response · Authors · 2026-04-07
> > >
> > > We thank the reviewer for the detailed follow-up and for clearly articulating the remaining concerns. We address each point below.
> > >
> > > > **W4-5, Q1-2:** Training rather than existence.
> > >
> > > We first clarify the position of IMDM. In this paper, we consider problems in few-step distillation of masked diffusion models (MDMs). The objective of few-step distillation is minimizing the factorization error $TC_\theta(z_s|z_t)$, which hinders few-step generation of discrete diffusion models (Sec. 3.2). Our theoretical finding, stated in Theorem 4.1, shows that standard MDM has an irreducible bound on factorization error regardless of model capacity or the choice of few-step distillation method. To remove this bound, we propose IMDM that introduces infinite mask categories to MDMs while being compatible with pretrained MDMs (Sec. 4.2). As shown in Theorem 4.2, IMDM admits an optimal $\theta^*$ that achieves zero factorization error. To reach the solution, IMDM is **not** trained with the masked prediction objective (Eq. 6); instead, with few-step distillation methods such as SDTT and ReDi, which can be interpreted as minimizing the total correlation (Eq. 7; L272-277). Under this objective, the model is driven toward noise-exploiting solutions, resulting in better few-step generation performance (Sec. 5).
> > >
> > > We note that the masked prediction objective (Eq. 6) and its results (Table 2) appear only to demonstrate that pretrained MDM weights transfer seamlessly to IMDM, and are not used for the few-step generation. We will make this distinction clearer in the revision.
> > >
> > > We now explain why the model learns to exploit noise. As discussed above, Theorem 4.1 establishes that any noise-ignoring solution (i.e., a standard MDM with deterministic [MASK]) has an irreducible factorization error. Theorem 4.2 (Appx. A.2) establishes that an optimal noise-exploiting solution achieves zero factorization error. Since the few-step distillation methods can be interpreted as minimizing the total correlation (Eq. 7), gradient descent with these methods has an incentive to move toward noise-exploiting solutions. This also explains why the zero-initialized MLP does not remain at zero: at initialization the model is subject to the irreducible error of Theorem 4.1, and updating the MLP offers lower factorization error.
> > >
> > > Regarding conflicting training signals (Q1), we note that such conflicts would require multiple conditions (matching context, timestep, masking pattern, and nearby noise) to hold simultaneously, which is highly unlikely in practice. We believe this is why training is stable.
> > >
> > > Lastly, regarding the inconsistency the reviewer identified: our previous statement that "IMDM does not learn a static mapping" was misleading. As the above explanation makes clear, IMDM does learn a deterministic noise-to-output mapping consistent with the construction in Theorem 4.2, conditioned on the full input. We apologize for the confusion.
> > >
> > > > Explanation on the diversity trend
> > >
> > > The decreasing diversity with more sampling steps is a general property of few-step distillation-based MDM sampling, not specific to IMDM. As the number of steps increases, each step unmasks fewer tokens, constraining each step's output to be more consistent with the already-unmasked context and reducing overall sample diversity. This trend is shared across all few-step distillation baselines in our experiments (e.g., SDTT: DIV 0.38 → 0.21 for 2 → 16 steps), and is also observed in Duo [1], where MDLM with SDTT shows sample entropy decreasing from approximately 5.6 to 5.4 as the number of steps increases (Figure 3).
> > >
> > > That said, we agree with the reviewer's concern that Gen.PPL improvement could in principle come from a narrower output distribution rather than better modeling. This is precisely why we provided the DIV-matched comparison in Tables A-B of our previous rebuttal. By adjusting IMDM's sampling temperature to match SDTT's and Di4C's DIV, we showed that IMDM achieves lower Gen.PPL at the same DIV level, indicating that the improvement stems from better modeling.
> > >
> > > [1] Sahoo et al. The Diffusion Duality. 2025.
> > >
> > > > Inconsistent results, reproducibility, and presentation.
> > >
> > > We apologize for the confusion regarding the inconsistent numbers. This discrepancy occurred because the original paper reported results from the running model, whereas our previous rebuttal used the EMA model. The corrected results are provided in our response to Reviewer N2UU (W2, Q2), and all conclusions of the paper remain the same. Regarding sampling details (W6), all reported results in the paper use standard categorical sampling without temperature, top-k, or top-p. We will state this explicitly and number the unnumbered equations (W7) in the revision.
> > >
> > > We hope these clarifications adequately address the reviewer's remaining concerns, and we would be grateful if the reviewer could consider updating their assessment in light of the above responses.

---

### Decision · Program_Chairs · 2026-04-30

**Decision:**

Accept (regular)

**Comment:**

There are mixed opinions about this work. Reviewers praised the theoretical analysis of the factorization error and the proof showing the existence of an optimal IMDM model that achieves zero error. The authors also successfully defended their novelty against baselines (e.g., VADD) during the rebuttal. One reviewer was not satisfied with the rebuttal due to concerns that the theoretical analysis does not guarantee reduced sampling error. The AC's assessment is that the main theoretical contribution is largely a straightforward extension of existing MDM theory based on total correlation. Therefore, the overall contribution feels marginal and I lean towards a weak accept.